# SAKE: Towards Editing Auditory Attribute Knowledge of Large Audio-Language Models

## Abstract

Knowledge editing offers an efficient way to update model knowledge without full retraining, but prior work has concentrated almost exclusively on textual or visual modalities. We introduce SAKE, the first benchmark specifically designed for editing auditory attribute knowledge in Large Audio-Language Models (LALMs). Unlike factual updates, SAKE targets several abstract auditory attributes, capturing knowledge types that go beyond conventional textual and visual domains. We benchmark eight editing methods on two LALMs along four dimensions: *reliability*, *generality*, *audio/text locality*, and *portability*. Results highlight challenges such as preserving intra-attribute knowledge unrelated to the edit, generalizing edits to multimodal reasoning, and maintaining edits under sequential updates. SAKE provides a principled framework to study how knowledge editing extends to the auditory modalities, opening new directions for maintaining and adapting LALMs in more diverse real-world scenarios.

## 1 Introduction

Large language models (LLMs) (Touvron et al., 2023; Grattafiori et al., 2024; Hurst et al., 2024) have demonstrated remarkable progress across natural language understanding and generation tasks. As they scale, knowledge editing (De Cao et al., 2021; Mitchell et al., 2022; Zheng et al., 2023; Deng et al., 2025) has emerged as an important line of research. The goal of knowledge editing is to efficiently update specific pieces of model knowledge without full retraining while minimizing the risk of catastrophic forgetting. These techniques enable incorporating new facts (De Cao et al., 2021), correcting errors, mitigating biases (Chen et al., 2024a), and supporting personalization (Lu et al., 2025d), making knowledge editing a key tool for adapting LLMs to real-world use.

Advances in LLMs have also led to multimodal extensions such as large vision-language models (LVLMs) (Liu et al., 2023; Li et al., 2023) and large audio-language models (LALMs) (Lu et al., 2025b; Chu et al., 2024; Kuan et al., 2024; Ghosh et al., 2025; Hurst et al., 2024; Lin et al., 2025; Lu et al., 2025a; Gong et al., 2023; Tang et al., 2024), which integrate additional modalities into LLMs. As multimodal models gain adoption, it becomes increasingly important to extend knowledge editing beyond the textual domain. Recent efforts have explored editing LVLMs through benchmarks targeting visual knowledge updates (Cheng et al., 2023; Huang et al., 2024; Zhang et al., 2024). However, knowledge editing in auditory modalities remains unexplored despite the rising importance of LALMs for speech and audio understanding.

Editing auditory attribute knowledge (Yang et al., 2025a) introduces unique challenges and opportunities. Factual knowledge (Thorne et al., 2018) typically involves concrete statements (e.g., "Paris is the capital of France"), that can be typically represented through explicit triplets of subjects, relations, and objects. In contrast, auditory attributes such as speaker gender, emotion, spoken language, or animal sounds represent high-level perceptual concepts. These attributes may manifest through infinitely many acoustic realizations, including different speakers, prosodic variations, or recording conditions, while still grounding the same underlying attribute. As a result, it is unclear whether existing methods developed for editing concrete factual knowledge can be extended to such abstract representations. At the same time, editing auditory attributes can extend applications in the textual domain, such as debiasing (Chen et al., 2024a) and personalization (Lu et al., 2025d), into the auditory modalities, with new possibilities such as mitigating gender biases in LALMs (Lin et al.,

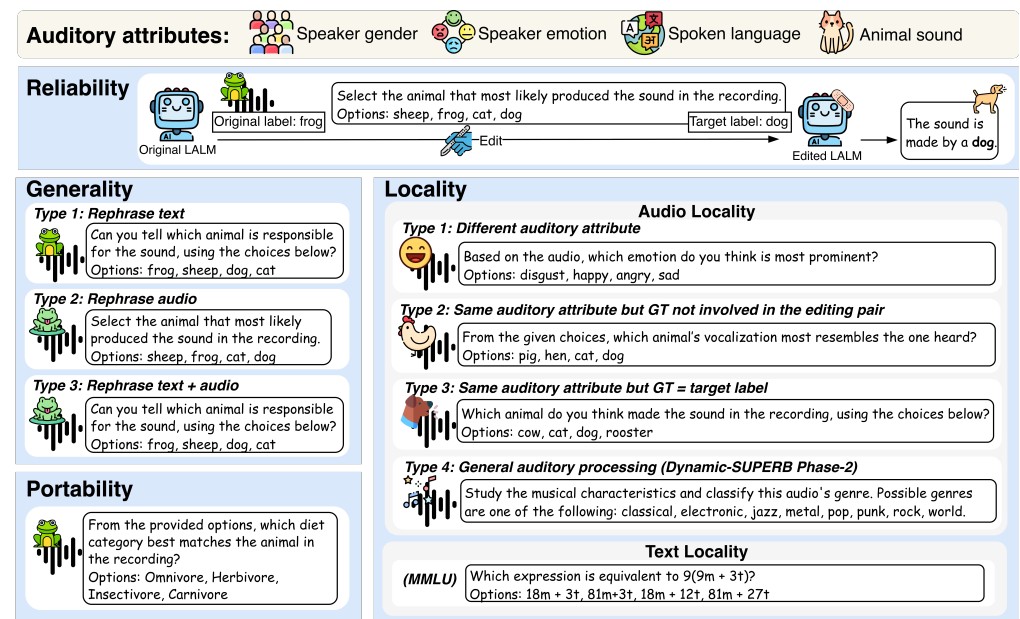

Figure 1: Overview of SAKE benchmark. Four auditory attributes are targeted. Reliability checks if the edit succeeds, Generality if it holds for equivalent data, Locality if unrelated knowledge remains unchanged, and Portability if it transfers to related knowledge. For example, after editing "frog" to "dog," the answer of the portability question should change from "Insectivore" to "Omnivore."

2024b) and enabling personalization (Yang et al., 2025c; Lee et al., 2024) by adapting to a user's unique voice or speaking style.

To this end, we introduce **SAKE** (**S**peech and **A**udio Attribute **K**nowledge **E**diting Benchmark), the first benchmark for auditory attribute knowledge editing in LALMs (Figure 1). SAKE covers four auditory attributes: speaker gender, speaker emotion, spoken language, and animal sounds, and evaluates editing methods along four dimensions (Yao et al., 2023): reliability (whether edits succeed), generality (whether they transfer to equivalent variations), locality (whether unrelated knowledge is preserved), and portability (whether edits propagate to interconnected knowledge).

Through comprehensive experiments on two strong LALMs, DeSTA2.5-Audio (Lu et al., 2025b) and Qwen2-Audio (Chu et al., 2024), we benchmark diverse editing techniques and reveal significant challenges in editing auditory attribute knowledge, such as preserving intra-attribute knowledge that is irrelevant to the edits and ensuring that updated auditory attributes propagate to reasoning over related knowledge. Moreover, when multiple edits are applied sequentially, most methods suffer from severe forgetting, often losing previously edited knowledge after a few additional updates. Overall, our contributions are two-fold[1]:

1. To the best of our knowledge, this is the first work to explore knowledge editing in auditory modalities of LALMs, an area that has remained unexplored. We introduce the first comprehensive benchmark for auditory knowledge editing and evaluate eight common editing methods, providing novel insights into this direction.

2. Although these editing methods are effective in textual and visual domains, editing auditory attributes, which are abstract perceptual concepts, remains challenging. We show that current methods struggle both to generalize edited knowledge and to preserve unrelated knowledge, underscoring the need for future advances in auditory knowledge editing.

---

[1]Resources will be available at `https://github.com/ckyang1124/SAKE`

## 2 RELATED WORK

### 2.1 KNOWLEDGE EDITING

Knowledge editing (Zhu et al., 2024) refers to techniques designed to efficiently update or modify the knowledge stored in models. The goal is to adjust model knowledge in a lightweight manner while avoiding catastrophic forgetting that may arise from retraining directly on the target knowledge. Existing methods (Mitchell et al., 2022; De Cao et al., 2021; Zheng et al., 2023; Meng et al., 2022) adopt various strategies: using a hypernetwork to predict parameter updates for incorporating new knowledge (De Cao et al., 2021; Mitchell et al., 2022), identifying and adjusting neurons associated with specific knowledge (Meng et al., 2022), or leveraging in-context learning (ICL) to enable models to acquire updated knowledge (Zheng et al., 2023). More recent works have extended the line of work from Meng et al. (2022) to mass editing (Meng et al., 2023) and unstructured scenarios (Deng et al., 2025; Jiang et al., 2025; Su et al., 2025). Beyond correcting factual knowledge, these techniques have also been applied to tasks such as bias mitigation (Chen et al., 2024a), detoxification (Wang et al., 2024a), personalization (Lu et al., 2025d), and unlearning (Li et al., 2025).

More recently, researchers have begun to investigate knowledge editing in large vision-language models (LVLMs)(Liu et al., 2023; Li et al., 2023). For example, Cheng et al. (2023) introduced the MMEdit benchmark to explore editing visual knowledge, while subsequent works such as VLKEB (Huang et al., 2024) and MC-MKE (Zhang et al., 2024) expanded the evaluation scope to provide a more comprehensive understanding of editing in visual modalities. However, no prior work has examined editing auditory attribute knowledge in LALMs, which involves abstract perceptual concepts rather than concrete facts, distinguishing our study from existing research.

### 2.2 LARGE AUDIO-LANGUAGE MODELS (LALMS)

LALMs extend text-based LLMs to auditory modalities such as speech and audio, opening new possibilities for auditory understanding (Gong et al., 2023; Tang et al., 2024; Chu et al., 2023; 2024; Lu et al., 2025b; Ghosh et al., 2025). These models typically integrate auditory encoders (Radford et al., 2023) with an LLM backbone (Touvron et al., 2023; Yang et al., 2024a) through fine-tuning. While these works advance the integration of auditory knowledge into text-based LLMs, little attention has been given to how auditory-specific knowledge can be edited or updated, which motivates our study.

## 3 SAKE: SPEECH & AUDIO ATTRIBUTE KNOWLEDGE EDITING BENCHMARK

### 3.1 PROBLEM FORMULATION

Given an LALM $f$ with parameters $\theta$ and an editing dataset $\mathcal{D}_{edit} = \{(a_e, x_e, y_e)\}$, where $a_e$ denotes the auditory input, $x_e$ the text input, and $y_e$ the desired edit target, knowledge editing aims to update the model such that the edited parameters $\theta'$ enable the LALM to faithfully generate the edit target: $f(a_e, x_e; \theta') = y_e$.

In this work, we focus on editing auditory attribute knowledge within LALMs, including their perception and understanding of speaker gender, emotion, spoken language, and animal sounds. In this setting, $y_e$ corresponds to new auditory attribute labels (e.g., emotions or languages) that differ from the original attribute labels $y_o$ associated with $a_e$. For example, given a speech labeled with a happy emotion, we may edit the LALM so that it instead perceives the speech as having a sad tone.

For comprehensive evaluation, we introduce the four evaluation dimensions of SAKE and the corresponding metrics, followed by dataset construction details for each dimension.

### 3.2 EVALUATION DIMENSIONS AND CORRESPONDING METRICS

We introduce the four dimensions of knowledge editing, namely reliability, generality, locality, and portability, together with their corresponding evaluation metrics.

**Reliability.** The **reliability** metric $S_{rel}$ measures the proportion of editing instances in $\mathcal{D}_{edit}$ for which the edited model correctly generates the corresponding edit target. It reflects how consistently the editing method updates the model in the desired manner, and is defined as

$$S_{rel} = \mathbb{E}_{(a_e, x_e, y_e) \sim \mathcal{D}_{edit}} \Big[ \mathbb{I}\big(f(a_e, x_e; \theta') = y_e\big) \Big], \tag{1}$$

where $\mathbb{I}$ denotes the indicator function, which returns 1 if the condition holds and 0 otherwise.

**Generality.** The edited models should not only generate the correct edit target for the editing data itself but also produce consistent outputs for equivalent neighborhoods of the editing data, such as speech samples sharing the same emotion as $a_e$ or paraphrased variants of $x_e$. This requirement is quantified by the **generality** metric $S_{gen}$, defined as

$$S_{gen} = \mathbb{E}_{\substack{(a_e, x_e, y_e) \sim \mathcal{D}_{edit} \\ (a'_e, x'_e) \sim \mathcal{N}(a_e, x_e)}} \Big[ \mathbb{I}\big(f(a'_e, x'_e; \theta') = y_e\big) \Big], \tag{2}$$

where $\mathcal{N}(a_e, x_e)$ denotes the aforementioned equivalent neighborhood of the editing data $(a_e, x_e)$.

**Locality.** While updating the edit target, the edit should also preserve unrelated knowledge to avoid unintended side effects. The **locality** metric $S_{loc}$ evaluates how well an editing method maintains the model's knowledge outside the editing scope. Given a set of out-of-scope data $\mathcal{L}(a_e, x_e, y_e) = \{(a_\ell, x_\ell, y_\ell)\}$, consisting of auditory inputs, text inputs, and ground-truth labels, $S_{loc}$ is defined as the proportion of out-of-scope data where the model's behavior remains unchanged after editing:

$$S_{loc} = \mathbb{E}_{\substack{(a_e, x_e, y_e) \sim \mathcal{D}_{edit} \\ (a_\ell, x_\ell, y_\ell) \sim \mathcal{L}(a_e, x_e, y_e)}} \Big[ \mathbb{I}\big(f(a_\ell, x_\ell; \theta') = f(a_\ell, x_\ell; \theta)\big) \Big]. \tag{3}$$

Note that the locality metric evaluates whether the post-edit model preserves the knowledge and behavior on data irrelevant to the edit, rather than the accuracy on out-of-scope instances. For locality with respect to purely textual abilities, we set $a_\ell = \text{None}$, as no auditory input is involved.

**Portability.** Knowledge is not completely disentangled or isolated but rather interconnected. Editing one piece of knowledge may influence other related knowledge. For example, if we edit an LALM's perception of a frog's sound to that of a dog, the model's knowledge of the corresponding physical characteristics of that animal should also be updated. The **portability** metric $S_{port}$ evaluates how well the edited model generalizes the updated knowledge to other related knowledge:

$$S_{port} = \mathbb{E}_{\substack{(a_e, x_e, y_e) \sim \mathcal{D}_{edit} \\ (a_p, x_p, y_p) \sim \mathcal{P}(a_e, x_e, y_e)}} \Big[ \mathbb{I}\big(f(a_p, x_p; \theta') = y_p\big) \Big]. \tag{4}$$

Here, $\mathcal{P}(a_e, x_e, y_e)$ denotes the set of data connected to the edited knowledge, with $a_p$, $x_p$, and $y_p$ representing the auditory input, text input, and ground-truth labels of these connected instances.

## 3.3 DATASET CONSTRUCTION

We introduce the **SAKE** benchmark to evaluate the knowledge editing methods on editing the auditory attribute knowledge in LALMs with respect to the metrics detailed in Sec. 3.2. We benchmark the editing methods on LALMs with speech and audio multiple-choice question answering.

**Involved Auditory Attribute Knowledge and Audio Sources.** SAKE focuses on editing the knowledge of four different auditory attributes: speaker gender, speaker emotion, spoken language, and animal sound. They are chosen for their importance in many downstream applications.

We source audio data from the SAKURA benchmark (Yang et al., 2025b), which evaluates LALMs on recognizing these four auditory attributes and related multi-hop reasoning. SAKURA compiles audio samples and attribute labels from CommonVoice (Ardila et al., 2020), CREMA-D (Cao et al., 2014), ESC-50 (Piczak, 2015), and the Animal-Sound Dataset (Şaşmaz & Tek, 2018). We also extract the auditory attribute labels from SAKURA for these attributes to form a set of labels[2].

---

[2]Gender: Male, Female; Language: English, German, Spanish, French, Italian, Chinese, Japanese, Korean; Emotion: Happy, Disgust, Sad, Fear, Angry; Animal: Dog, Cat, Pig, Cow, Frog, Hen, Rooster, Sheep, Crow.

**Editing Pair Creation.** We begin by constructing editing pairs $(y_o, y_e)$, where $y_o$ is the original attribute label and $y_e$ the target label after editing. For each attribute, we generate 300 editing pairs by uniformly sampling one label as $y_o$ and another distinct label as $y_e$. For example, the editing pair $(\mathrm{dog}, \mathrm{cat})$ represents an edit in which the model's perception and understanding of dog sounds are updated to those of cat sounds. To avoid bias, we ensure that all labels for a given attribute appear with approximately equal frequency as both $y_o$ and $y_e$.

**Reliability Dataset Construction.** For each attribute and a given editing pair $(y_o, y_e)$, we sample an audio instance $a_e$ with label $y_o$ from the dataset, along with the corresponding text questions $x_e$ from the SAKURA benchmark that prompt the model to recognize the attribute. Together with the edit target $y_e$, this forms an editing instance $(a_e, x_e, y_e)$. For each attribute, we construct 300 such instances, resulting in a total of 1,200 editing instances. This dataset, denoted as $\mathcal{D}_{edit}$, is used both for applying the edits and for evaluating the reliability of the editing methods.

**Generality Dataset Construction.** For each editing instance $(a_e, x_e, y_e)$ in $\mathcal{D}_{edit}$, we construct its equivalent neighborhood $\mathcal{N}(a_e, x_e) = (a'_e, x'_e)$ by sampling an alternative audio $a'_e$ with the same attribute label as $a_e$ from the dataset and by paraphrasing the text question $x_e$ into $x'_e$. All the paraphrased data are manually verified. Based on these variations, we create testing instances for evaluating generality, considering three types of cases: **Type 1**: Equivalent neighborhood of the text modality $(a_e, x'_e)$; **Type 2**: Equivalent neighborhood of the auditory modality $(a'_e, x_e)$; **Type 3**: Equivalent neighborhood involving both auditory and text modalities, resulting in instances of the form $(a'_e, x'_e)$. By incorporating these three types of generality testing data, we comprehensively assess how well the editing methods extend the edited knowledge across the equivalent neighborhood.

**Locality Dataset Construction.** Similar to the construction of the generality dataset, for each editing instance $(a_e, x_e, y_e)$ in $\mathcal{D}_{edit}$, we construct an out-of-scope dataset $\mathcal{L}(a_e, x_e, y_e) = \{(a_\ell, x_\ell, y_\ell)\}$. When $a_\ell \neq$ None, we consider **four** types of knowledge locality associated with auditory modalities, which we refer to as Audio Locality in this paper, as illustrated in Figure 1. First, editing one attribute should not affect others. We sample a question-answer pair from the SAKURA benchmark that requires the LALM to recognize attributes **different from** those in the editing instance as $(a_\ell, x_\ell, y_\ell)$. Second, even within the same attribute, labels that are not involved in the edit (i.e., neither $y_o$ nor $y_e$) should not be perturbed. Accordingly, we sample a question-answer pair from SAKURA requiring recognition of **the same** attribute as the editing instance, but with ground truth differing from both $y_o$ and $y_e$. Note that for the gender attribute, which only has two labels, it is infeasible to construct such cases; therefore, this type of locality data is not included for gender. Third, when editing from $y_o$ to $y_e$, the model's original knowledge of $y_e$ should be preserved, which we test with a question-answer pair from SAKURA that requiring recognizing **the same** attribute as the editing instance, with ground truth $y_e$. Finally, editing should not interfere with general auditory processing. We source question-answer pairs from Dynamic-SUPERB Phase-2 (Huang et al., 2025), a benchmark covering diverse auditory processing tasks. We exclude tasks involving the four attributes considered in our work to ensure irrelevance to the edited knowledge. Conversely, to assess the preservation of purely text-based knowledge when $a_e =$ None (text locality), we use question–answer data from MMLU (Hendrycks et al., 2021).

**Portability Dataset Construction.** For each editing instance $(a_e, x_e, y_e)$ in $\mathcal{D}_{edit}$, we construct a set of connected knowledge $\mathcal{P}(a_e, x_e, y_e)$ associated with the edited attribute. Since SAKURA is designed to evaluate how well LALMs integrate internal world knowledge with auditory attribute knowledge, it already provides relevant knowledge linked to auditory attributes (e.g., physical characteristics of animal labels). Building on this, we create questions that specifically target $\mathcal{P}(a_e, x_e, y_e)$. To avoid ambiguity, we ensure that the answers to these questions are not simultaneously valid for both $y_o$ and $y_e$. For example, if $y_o$ is dog and $y_e$ is cat, a question about the animal's physical characteristics will not use an answer like "tail," which applies to both. This guarantees that the knowledge examined in the portability dimension indeed requires updating after the edit.

**Training Dataset Construction and the Dataset Summary.** We also prepare a separate training dataset to accommodate editing methods that require (1) training on auxiliary data or (2) access to additional data beyond the testing instances. Its construction follows the same procedures as those used for the reliability, generality, and locality datasets, with training and testing sets kept fully disjoint to avoid leakage. In total, the training set contains 4,000 instances (32,000 speech/audio files and 36,000 QA pairs), while the testing set comprises 1,200 instances (10,800 speech/audio files and 12,000 QA pairs). A summary of dataset statistics is provided in Appendix B.

# 4 EXPERIMENTAL SETTINGS

We experiment on two LALMs, DeSTA2.5-Audio (Lu et al., 2025b) and Qwen2-Audio-Instruct (Chu et al., 2024), chosen for their strong benchmark performance (Huang et al., 2025; Yang et al., 2025b; Sakshi et al., 2025; Chen et al., 2024b; Lu et al., 2025c). We apply greedy decoding and assess editing methods under two settings: single editing and sequential editing.

## 4.1 EDITING METHODS

We evaluate eight editing methods, focusing on their effectiveness in modifying abstract auditory attribute knowledge. Below, we briefly introduce these methods, with the details provided in Appendix D.

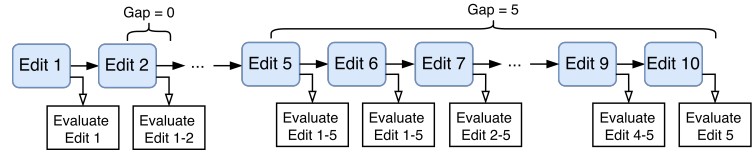

Figure 2: Example of sequential editing. For comparability, only the first five edits are evaluated, with gaps capped at five. For example, evaluating Edit 2 immediately results in a gap of 0, whereas after Edit 10, only Edit 5 can be evaluated, with a gap of 5 under this rule.

**Fine-tuning** is a common approach for adapting pre-trained models to new knowledge. Following prior work on LVLMs (Cheng et al., 2023; Huang et al., 2024; Zhang et al., 2024), we compare fine-tuning two parts in LALMs: the last layer of the LLM backbone, denoted as **FT (LLM)**, and the modality connector between the audio encoder and the LLM backbone, denoted as **FT (Audio)**. **Knowledge Editor (KE)** (De Cao et al., 2021) and **MEND** (Mitchell et al., 2022) trains a hyper-network to transform the fine-tuning gradients into parameter updates for the edit. **UnKE** (Deng et al., 2025) optimizes specific neurons of the chosen layers to produce the edit target. **In-Context Knowledge Editing (IKE)** (Zheng et al., 2023) leverages in-context learning (ICL) to enforce knowledge updates. We consider two variants: **Instruction-based IKE (I-IKE)**, which encodes edits solely through natural language instructions in system prompts, and **Instruction+Example IKE (IE-IKE)**, which provides auditory examples retrieved from the training set as dialog context. **WISE** (Wang et al., 2024b) edits models using a dual-memory design that keeps pretrained FFN weights intact while storing edit-specific updates in a lightweight side memory.

## 4.2 SINGLE EDITING AND SEQUENTIAL EDITING

**Single editing** evaluates the performance of editing a single piece of knowledge, whereas **sequential editing** evaluates the performance after applying a sequence of edits continuously on different knowledge, which better reflects real-world scenarios.

For sequential editing, we construct ten independent sequences, each with ten editing instances. An editing sequence is denoted as $\mathcal{S} = \{(a_e^{(t)}, x_e^{(t)}, y_o^{(t)}, y_e^{(t)})\}_{t=1}^{10}$, where $(a_e^{(t)}, x_e^{(t)}, y_e^{(t)})$ is sampled from $\mathcal{D}_{edit}$ and the corresponding original label $y_o^{(t)}$ is retrieved from our audio dataset. Here, $t$ indexes the order of edits within the sequence.

To measure how long an edit remains effective under subsequent edits, we define the *gap* as the number of editing steps between when an edit is applied and when it is evaluated (Figure 2). If an edit is introduced at step $j$ and evaluated at step $i$, then the gap is $i - j$. For consistency, we only consider the first five edits and require the gap to be at most five. This restriction keeps the number of samples comparable across gaps, since larger gaps naturally yield fewer available evaluations. To guarantee the validity of edit sequences, we impose two rules when sampling: (1) **Editing pair independence:** All original and edited labels in the sequence are mutually distinct, i.e., $y_o^{(1)}, y_e^{(1)}, \ldots, y_o^{(10)}, y_e^{(10)}$ appear only once within the sequence. This avoids contradictions that could compromise the evaluation of edits with subsequent ones (e.g., editing "dog sounds" to "cat sounds" and later editing "dog sounds" to "frog sounds"); and (2) **Audio locality independence:** For each sequential editing instance $(a_e^{(t)}, x_e^{(t)}, y_o^{(t)}, y_e^{(t)})$, all samples in its audio locality dataset $\mathcal{L}(a_e^{(t)}, x_e^{(t)}, y_e^{(t)})$, denoted

Table 1: The four metrics (%) of the editing methods on the two models. Avg. indicates the average performance across all types of the corresponding metric. The best and second-best results are highlighted in **bold** and underlined, respectively.

| | Method | Reliability | Generality | | | | Audio Locality | | | | | Text Locality | Portability |
|---|---|---|---|---|---|---|---|---|---|---|---|---|---|
| | | | Avg. | Type 1 | Type 2 | Type 3 | Avg. | Type 1 | Type 2 | Type 3 | Type 4 | | |
| **DeSTA2.5-Audio** | **FT (LLM)** | 99.75 | 98.75 | 99.67 | 99.00 | 97.58 | 65.11 | 88.08 | 15.56 | 74.67 | 69.75 | 82.58 | 19.42 |
| | **FT (Audio)** | 99.50 | 86.14 | 96.75 | 84.92 | 76.75 | 68.09 | 78.67 | 48.11 | 76.17 | 64.42 | **100.00** | 55.50 |
| | **KE** | 99.58 | 99.17 | 99.25 | 99.25 | 99.00 | **79.29** | **96.33** | 43.89 | 76.33 | **91.75** | 92.50 | 18.42 |
| | **MEND** | 95.33 | 95.00 | 95.83 | 95.17 | 94.00 | 71.47 | 93.42 | 17.22 | 74.67 | 87.00 | 92.08 | 19.42 |
| | **UnKE** | 96.33 | 89.28 | 89.67 | 94.25 | 83.92 | 56.04 | 67.67 | 10.56 | 71.92 | 62.67 | 87.42 | 18.75 |
| | **I-IKE** | 73.00 | 61.47 | 64.67 | 59.58 | 60.17 | 65.40 | 79.42 | **50.67** | 73.67 | 54.17 | 62.08 | **71.42** |
| | **IE-IKE** | 40.58 | 39.61 | 41.50 | 38.67 | 38.67 | 58.76 | 70.08 | 49.89 | 70.75 | 42.08 | 56.25 | 34.58 |
| | **WISE** | **100.00** | **100.00** | **100.00** | **100.00** | **100.00** | 37.00 | 27.85 | 3.22 | 76.00 | 26.62 | 70.25 | 11.40 |
| **Qwen2-Audio** | **FT (LLM)** | **100.00** | **99.94** | **100.00** | **100.00** | **99.83** | 67.42 | 91.67 | 10.44 | 83.33 | 70.00 | 74.58 | 24.67 |
| | **FT (Audio)** | **100.00** | 81.86 | 99.00 | 77.17 | 69.42 | **90.53** | 96.83 | 80.11 | 90.00 | **92.58** | **100.00** | **50.67** |
| | **KE** | 95.50 | 86.67 | 92.00 | 87.67 | 80.33 | 83.47 | 89.83 | 61.44 | 87.25 | 89.83 | 84.92 | 27.58 |
| | **MEND** | **100.00** | 95.33 | 98.92 | 95.92 | 91.17 | 83.27 | **98.50** | 49.33 | 85.42 | 91.33 | 86.75 | 27.17 |
| | **UnKE** | 98.58 | 98.53 | 99.08 | 98.83 | 97.67 | 67.49 | 91.42 | 12.34 | 82.92 | 69.50 | 71.58 | 28.42 |
| | **I-IKE** | 10.33 | 7.11 | 10.33 | 5.67 | 5.33 | 87.51 | 94.75 | **89.33** | **93.00** | 73.42 | 55.00 | 28.92 |
| | **IE-IKE** | 8.00 | 6.58 | 8.50 | 5.75 | 5.50 | 82.89 | 91.25 | 86.56 | 89.67 | 65.00 | 50.00 | 27.50 |
| | **WISE** | **100.00** | 99.31 | 99.83 | 99.17 | 98.92 | 69.02 | 92.42 | 5.67 | 82.67 | 74.67 | 64.50 | 27.00 |

by $(a_\ell^{(t)}, x_\ell^{(t)}, y_\ell^{(t)})$ where $a_\ell^{(t)} \neq$ None, are unrelated to the original labels of all edited instances $y_o^{(1..10)}$, thereby ensuring independent evaluation of the current edit.

### 4.3 EVALUATOR

Because LALMs often generate descriptive responses, we adopt LLM-as-a-judge (Chiang & Lee, 2023) to compute the metrics introduced in Sec. 3.2. In particular, we employ GPT-5 mini (`gpt-5-mini-2025-08-07`) as the evaluator, focusing on *correctness* and *consistency*.

For the reliability, generality, and portability metrics, we assess whether the edited model's responses correctly align with the ground truth. For locality, we examine whether the edited model preserves consistency with the original model's outputs. To further validate the quality of the LLM-based judgments, we conduct human evaluation on 420 randomly selected samples. The results show overall 98.10% agreement with the LLM evaluator, demonstrating its robustness. Additional details and the evaluation prompts are provided in Appendix E.

## 5 RESULTS

### 5.1 SINGLE EDITING

The main results of different knowledge editing methods on SAKE are shown in Table 1. Detailed results for each auditory attribute and relevant discussions are provided in Appendix F.

**Reliability.** For both LALMs, most editing methods achieve high reliability, with WISE consistently yielding the best scores. In contrast, I-IKE and IE-IKE perform poorly, reaching only 73.00% and 40.58% on DeSTA2.5-Audio, and 10.32% and 8.00% on Qwen2-Audio. Interestingly, I-IKE outperforms IE-IKE on both models despite the latter using additional auditory examples. We attribute this to LALMs' limited in-context learning ability: they struggle to handle multi-audio inputs and leverage examples, unlike in LLMs (Zheng et al., 2023) and LVLMs (Cheng et al., 2023; Huang et al., 2024) where such methods are effective.

**Generality.**   Compared with reliability, editing methods yield lower average generality scores. WISE performs best on DeSTA2.5-Audio, while FT (LLM) leads on Qwen2-Audio. Baselines generally perform well on type 1 (textual neighborhood) but decline on type 2, which requires generalizing to similar but not identical auditory inputs with the same labels. Type 3, combining both text and auditory neighborhoods, proves most challenging, showing that current methods struggle to extend edited knowledge to auditory modalities as effectively as to text.

Interestingly, FT (LLM) consistently outperforms FT (Audio) on generality despite their comparable reliability scores. This suggests that while fine-tuning either the LLM backbone or the modality connector ensures success on the editing instances themselves, the ability to generalize to similar inputs differs substantially, with training on the LLM backbone offering better generalizability.

**Audio Locality.**   Most methods perform poorly, with only limited success: KE preserves knowledge best on DeSTA2.5-Audio, while FT (Audio) performs better on Qwen2-Audio. This shows that maintaining stability in the auditory domain remains challenging despite advances in model editing.

Among audio locality types, type 2 locality, where knowledge within the same attribute but unrelated to the edit is examined, is hardest to preserve, more so than knowledge of other attributes (type 1) or the edit target itself (type 3). This suggests that edits on one aspect of an attribute may inadvertently affect other aspects of the same attribute, reflecting a degree of intra-attribute entanglement of auditory attribute knowledge. Notably, FT (LLM) performs much worse than FT (Audio) on type 2, indicating that despite better generalization, it fails to retain unrelated knowledge within the same attribute. By contrast, knowledge across different attributes (i.e., type 1) is less affected, indicating that cross-attribute boundaries tend to be more stable to preserve during editing. We provide a more detailed discussion of this phenomenon in Appendix H.

Focusing on type 4, which evaluates the preservation of general auditory processing capability, most editing methods struggle to retain LALMs' general abilities, suggesting a trade-off between effective editing and maintaining broad competence. Generally speaking, KE and MEND perform best, likely due to their regularization that emphasizes locality in hypernetwork training.

Although I-IKE and IE-IKE do not achieve high reliability and generality, they still preserve audio locality to some extent. This can be attributed to the limited in-context learning ability of DeSTA2.5-Audio and Qwen2-Audio, which reduces the extent to which editing propagates. However, since DeSTA2.5-Audio possesses stronger instruction-following capability (Lu et al., 2025b), it is more susceptible to perturbations during editing, leading to lower locality scores than Qwen2-Audio.

**Text Locality.**   FT (Audio) achieves 100% since it fine-tunes only the modality connector, which is triggered solely by auditory inputs and leaves text-only capabilities intact. FT (LLM) tunes the LLM backbone, impairing text-only tasks and yielding much lower scores. KE and MEND preserve text locality better through regularization on irrelevant samples during hypernetwork training. UnKE and WISE disrupt text-based knowledge, especially on Qwen2-Audio, while IKE variants also struggle, likely due to sensitivity to contextual information in text-only scenarios, even when irrelevant.

**Portability.**   Overall, current editing methods do not guarantee portability when modifying auditory attribute knowledge. Parameter-updating approaches generally overlook this aspect, which poses a major challenge for real-world deployment, where extensive related knowledge may need to be updated simultaneously. Among these methods, FT (Audio) achieves the most balanced performance on both LALMs, suggesting that training the modality connector can better integrate edited auditory knowledge with the internal world knowledge encoded in the LLM backbone.

Interestingly, I-IKE performs best on DeSTA2.5-Audio. This may be because portability is closely tied to reasoning, and the IKE variants benefit when the underlying model has sufficient reasoning proficiency. Since DeSTA2.5-Audio demonstrates substantially stronger reasoning ability than Qwen2-Audio (Lu et al., 2025b), the IKE variants perform better on it. However, due to DeSTA2.5-Audio's still-limited multi-audio processing ability, IE-IKE remains less effective than I-IKE.

**Summary.**   In sum, while most editing methods can adjust LALMs to produce the desired knowledge on edited instances, they struggle to generalize these changes to equivalent cases, especially for auditory inputs. They also struggle to extend edits consistently to interconnected knowledge. In addition, they often fail to preserve original knowledge within the same attribute, which highlights the

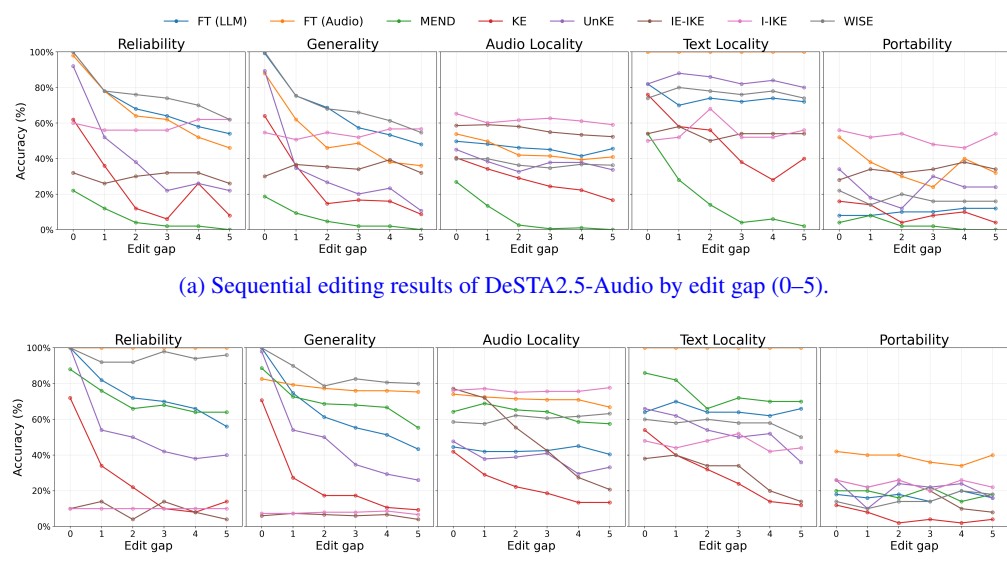

(a) Sequential editing results of DeSTA2.5-Audio by edit gap (0–5).

(b) Sequential editing results of Qwen2-Audio by edit gap (0–5).

Figure 3: Comparison of sequential editing results across models by edit gap (0–5).

difficulty of preventing unintended perturbations. Among existing methods, fine-tuning, especially FT (Audio), remains a strong baseline across our evaluations. Editing auditory attribute knowledge, therefore, remains highly challenging and calls for methods tailored to auditory modalities.

## 5.2 SEQUENTIAL EDITING

Figure 3a and 3b show sequential editing results on DeSTA2.5-Audio and Qwen2-Audio.

**Reliability and Generality.** Most methods decline in reliability and generality as the gap increases, indicating a tendency to forget previously edited auditory knowledge. On DeSTA2.5-Audio, MEND and KE deteriorate rapidly after a few edits, often degenerating into repetitive outputs, with examples provided in Appendix J. In contrast, although weaker in single-editing settings, the IKE variants remain comparatively stable across larger gaps, yielding stronger long-term reliability and generality on DeSTA2.5-Audio, particularly for I-IKE. WISE maintains good performance in sequential edit by isolating each update into its own masked subspace, preventing interference.

**Locality.** Most methods do not show the sharp declines seen in reliability and generality, suggesting sequential setups have a weaker impact on knowledge preservation. However, as discussed in Sec. 5.1, their robustness still leaves considerable room for improvement, as none of the methods fully address the challenge of consistent knowledge preservation. Notable exceptions are KE and MEND on DeSTA2.5-Audio and KE and IE-IKE on Qwen2-Audio, which show marked declines due to degeneration issues and the models' limited in-context learning ability.

**Portability.** Similarly, the portability metric does not show significant declines under sequential editing. I-IKE and FT (Audio) consistently outperform other methods on DeSTA2.5-Audio and Qwen2-Audio, respectively, in line with the single-edit results in Table 1. This suggests that their ability to update interconnected knowledge remains relatively stable across several edits. Nevertheless, the overall performance of all methods remains unsatisfactory, underscoring the need for further improvement.

## 6 CONCLUSIONS

In this paper, we present the first study on editing auditory attribute knowledge in LALMs. We introduce SAKE, a benchmark designed to evaluate this type of knowledge editing across four dimensions. Through comprehensive experiments applying eight representative methods to two state-of-the-art LALMs, we reveal limitations in preserving non-target auditory knowledge and in generalizing edits to interconnected world knowledge during reasoning. Our findings provide new

insights into this unexplored direction and establish a foundation for future research on developing more robust and specialized knowledge editing methods for auditory modalities.

**Limitations.** As the first benchmark for editing auditory attributes in LALMs, we focus on four representative attributes, though expanding further would allow broader evaluation. Similarly, while using two models follows common practice (Cheng et al., 2023; Zhang et al., 2024), including more LALMs could yield deeper insights. In addition, while our study focuses on speech-to-text LALMs, future work could extend to speech-to-speech LALMs (Fang et al., 2025; Xie & Wu, 2024; Yang et al., 2024b; Lin et al., 2024a; Chiang et al., 2025), which are likely to present greater challenges. Finally, we note that integrating interpretability analyses of the internal mechanisms of auditory knowledge editing could yield deeper insight into how the editing truly works. We leave these directions for future work.

## ETHICS STATEMENT

This work studies knowledge editing in large audio-language models (LALMs). Our datasets are publicly available and licensed for research. While knowledge editing offers benefits such as correcting errors and mitigating bias, we acknowledge potential misuse (e.g., manipulation of model knowledge) and emphasize that our contributions are intended for socially responsible applications. We report our methods and results transparently to ensure reproducibility.

## REPRODUCIBILITY STATEMENT

Our implementation of editing methods is built on the EasyEdit toolkit (Wang et al., 2024c), with hyperparameters provided in Appendix D.2. To reduce randomness in the LLM-as-a-judge evaluation, we fix the temperature at 0. We also report the version of the judge model and include the evaluation prompts in Appendix E. Together, these details help ensure the reproducibility.

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

# A  THE USAGE OF LARGE LANGUAGE MODELS (LLMs)

In this work, LLMs were used as judge models and as auxiliary tools for linguistic assistance, including polishing writing style, refining grammar, and proofreading. The conceptualization of the research problem, the design and construction of the benchmark, the execution of experiments, and the analysis and interpretation of results were carried out entirely by the authors without LLM involvement. All technical contributions and intellectual efforts originate from the authors, with LLMs serving only to support evaluation and to improve the clarity and readability of the manuscript.

# B  DATASET STATISTICS

We report the dataset statistics in Table 2 and Table 3, respectively, for the training and testing datasets, highlighting the diversity of the data.

Table 2: Dataset summary for each evaluation metric in our training dataset.

(a) Question Length (word).

| Metric | | Avg. | Min. | Max. | Std. |
|---|---|---|---|---|---|
| Reliability | | 16.40 | 12 | 22 | 2.32 |
| | Avg. | 16.40 | 12 | 22 | 2.32 |
| Generality | Type 1 | 16.40 | 12 | 22 | 2.32 |
| | Type 2 | 16.40 | 12 | 22 | 2.32 |
| | Type 3 | 16.40 | 12 | 22 | 2.32 |
| | Avg. | 26.03 | 8 | 604 | 57.53 |
| Audio | Type 1 | 16.40 | 12 | 22 | 2.32 |
| Locality | Type 2 | 16.90 | 13 | 22 | 2.23 |
| | Type 3 | 16.40 | 12 | 22 | 2.32 |
| | Type 4 | 52.13 | 8 | 604 | 107.09 |
| Text Locality | | 55.32 | 7 | 526 | 48.91 |

(b) Audio/speech duration (s).

| Metric | | Avg. | Min. | Max. | Std. |
|---|---|---|---|---|---|
| Reliability | | 4.62 | 0.13 | 17.81 | 2.27 |
| | Avg. | 4.62 | 0.13 | 17.81 | 2.27 |
| Generality | Type 1 | 4.62 | 0.13 | 17.81 | 2.27 |
| | Type 2 | 4.62 | 0.13 | 17.81 | 2.27 |
| | Type 3 | 4.62 | 0.13 | 17.81 | 2.27 |
| | Avg. | 8.27 | 0.11 | 1478.35 | 31.23 |
| Audio | Type 1 | 4.62 | 0.13 | 17.81 | 2.27 |
| Locality | Type 2 | 4.28 | 0.13 | 17.81 | 2.22 |
| | Type 3 | 4.62 | 0.13 | 17.81 | 2.27 |
| | Type 4 | 18.56 | 0.11 | 1478.35 | 59.16 |

Table 3: Dataset summary for each evaluation metric in our testing dataset.

(a) Question Length (word).

| Metric | | Avg. | Min. | Max. | Std. |
|---|---|---|---|---|---|
| Reliability | | 24.23 | 7 | 48 | 10.24 |
| | Avg. | 23.83 | 7 | 48 | 9.99 |
| Generality | Type 1 | 23.63 | 7 | 48 | 9.85 |
| | Type 2 | 24.23 | 7 | 48 | 10.24 |
| | Type 3 | 23.63 | 7 | 48 | 9.85 |
| | Avg. | 31.51 | 7 | 604 | 55.81 |
| Audio | Type 1 | 23.93 | 7 | 48 | 9.91 |
| Locality | Type 2 | 25.35 | 12 | 48 | 9.61 |
| | Type 3 | 23.94 | 7 | 48 | 9.92 |
| | Type 4 | 51.29 | 9 | 604 | 104.30 |
| Text Locality | | 53.59 | 7 | 307 | 42.28 |
| Portability | | 33.53 | 13 | 63 | 12.38 |

(b) Audio/speech duration (s).

| Metric | | Avg. | Min. | Max. | Std. |
|---|---|---|---|---|---|
| Reliability | | 4.75 | 0.24 | 20.78 | 2.43 |
| | Avg. | 5.82 | 0.24 | 20.78 | 4.98 |
| Generality | Type 1 | 4.75 | 0.24 | 20.78 | 2.43 |
| | Type 2 | 6.35 | 0.24 | 20.78 | 5.79 |
| | Type 3 | 6.35 | 0.24 | 20.78 | 5.79 |
| | Avg. | 8.12 | 0.17 | 759.53 | 26.98 |
| Audio | Type 1 | 4.78 | 0.28 | 20.78 | 2.46 |
| Locality | Type 2 | 4.41 | 0.24 | 20.78 | 2.38 |
| | Type 3 | 4.73 | 0.24 | 20.78 | 2.50 |
| | Type 4 | 17.63 | 0.17 | 759.53 | 50.89 |
| Portability | | 4.75 | 0.24 | 20.78 | 2.43 |

# C  CHOICES OF MODELS & COVERED ATTRIBUTES

## C.1  MODELS

We selected Qwen2-Audio (Chu et al., 2024) and DeSTA2.5-Audio (Lu et al., 2025b) because they are both strong, widely recognized, and representative open-source LALMs. Qwen2-Audio is broadly adopted across reasoning, safety, interpretability, and multimodal studies, making it a standard community baseline. DeSTA2.5-Audio is a newer model that achieves leading performance on

benchmarks such as SAKURA (Yang et al., 2025b) and MMAU (Sakshi et al., 2025), representing the latest progress in LALMs. Our focus is to evaluate editing methods, not to compare LALMs themselves, and it is standard practice to use a small number of representative models as the test bed. Moreover, these two models differ substantially in backbone, modality adaptation, and training pipeline. The fact that our observations appear consistently across both models and across many editing methods suggests that these challenges arise from limitations of current editing techniques, rather than from the particular models chosen.

## C.2 ATTRIBUTES

We chose these four attributes (i.e., animal sound, speaker emotion, speaker gender, spoken language) because they are fundamental to speech and audio understanding, widely used in real-world applications, and consistently evaluated in existing LALM benchmarks (Sakshi et al., 2025; Yang et al., 2025b; Huang et al., 2025). Since this is the first study on auditory knowledge editing, focusing on core attributes provides a clear and representative starting point. In addition, these attributes have comparatively rich resources including audio datasets and QA pairs, which is essential for constructing reliable evaluations, especially for our portability track that requires multi-hop reasoning. While future work may expand to additional auditory attributes, the current selection offers a focused and principled foundation for the first benchmark in this area.

# D IMPLEMENTATION DETAILS OF THE EDITING METHODS

## D.1 EDITING METHODS

**Fine-tuning** adapts the model via gradient descent on selected components. We apply it to the last layer of the LLM backbone and the modality connector between audio encoders and the backbone in LALMs.

**Knowledge Editor (KE)** (De Cao et al., 2021) employs a hyper-network to update parameters. It leverages a bidirectional LSTM with constrained optimization to predict weight updates.

**MEND** (Mitchell et al., 2022) uses a hypernetwork to generate parameter updates by decomposing fine-tuning gradients into low-rank forms and transforming them into parameter updates.

**UnKE** (Deng et al., 2025) is an unstructured knowledge editing method. UnKE first finds a modified key vector by adding a small residual (delta) to the hidden state of a chosen layer so that the model's output shifts to the desired target. In the second stage, the parameters of the chosen layer are updated to make the chosen layer naturally produce this new key vector.

**In-Context Knowledge Editing (IKE)** (Zheng et al., 2023) uses in-context learning (ICL) to modify model knowledge without parameter updates, relying on instructions and demonstrations to enforce the edited knowledge. We evaluate two variants: **Instruction-based IKE (I-IKE)** and **Instruction+Example IKE (IE-IKE)**.

**WISE** (Wang et al., 2024b) performs knowledge editing by introducing a dual-memory architecture that separates the pretrained FFN weights from a lightweight side-memory used to store edit-specific updates. During editing, WISE optimizes a routing-aware activation loss that forces edited queries to rely on the side-memory while keeping irrelevant inputs mapped to the original parameters, ensuring strong locality without degrading pretrained behavior.

## D.2 IMPLEMENTATION DETAILS

The hyperparameters of each editing method are shown in Table 4, and their approximate execution time on NVIDIA H100 GPU is reported in Table 5. For FT (LLM) and FT (Audio), early stopping is applied when the loss falls below 1e-2. For KE and MEND, which require training a hyper-network, we train the hyper-network for one epoch due to its rapid convergence on the validation set. For UnKE, the editing process consists of two stages with separate hyperparameters. The first stage uses `v_step` iterations to find the desired key vector, with its own learning rate `v_LR`. The second stage uses `optim_step` iterations to update the weights of the chosen layer so that it naturally produces the new key vector, controlled by a separate learning rate `optim_LR`. To prevent

interference with unrelated knowledge, stage 2 also incorporates preservation data. Specifically, we use 20 preservation samples for text and 5 for audio. For I-IKE and IE-IKE, the edited knowledge is encoded through natural language instructions in the system prompt, using the prompt template provided below. In IE-IKE, the demonstration examples are retrieved from the training set according to the editing pair. For example, when editing the model's perception of speaker emotion from happy to sad, we retrieve an editing instance from the training set involving the same (happy, sad) editing pair. The demonstrating examples cover reliability, generality, and locality. In WISE, $\alpha$ bounds locality activation, with smaller values enforcing stricter preservation. $\beta$ sets a minimum edit activation, where larger values drive more aggressive edits. $\gamma$ enforces a margin ensuring edit activations exceed locality activations. Finally, `act_ratio` scales the inference routing threshold: higher values favor original weights, while lower values route more queries to the side-memory.

---

**System Prompt Template for I-IKE and IE-IKE**

You are performing in-context knowledge editing, but only for questions about audio inputs.

From now on, when analyzing any audio clip, you must apply the following edits consistently:

EDITS (New facts):
{edits}

where each entry has the form 'pre_edit ' → 'post_edit '.

Rules:
1. If your reasoning or prediction about an audio clip would normally lead to 'pre_edit ', you must instead treat it as 'post_edit '.
2. All properties, attributes, and facts that belong to 'post_edit ' must be applied consistently, as if the audio were actually from 'post_edit '.
3. If the user's question is unrelated to these edits, you should answer normally without making changes.
4. Always ensure your final answers are fully consistent with the edited mapping.

---

# E   MORE DETAILS AND EVALUATION PROMPTS FOR LLM-AS-A-JUDGE

We use GPT-5 mini with minimal reasoning effort as the judge in our LLM evaluator, balancing strong alignment with human judgments and cost efficiency. The temperature is set to 0 to ensure maximal reproducibility. Human verification was conducted on 210 randomly selected samples for correctness and consistency respectively, yielding 98.57% and 97.62% agreement. As described in Sec. 4.3, evaluation tasks are categorized into correctness and consistency, with prompts provided below. To further reduce costs, some cases are judged directly without invoking the LLM evaluator: outputs that are empty are marked as incorrect/inconsistent, and responses that exactly match the ground truth (in correctness) or the original model's output (in consistency) are marked as correct/-consistent.

Table 4: Hyper-parameters of each editing method. I-IKE and IE-IKE are excluded because they do not modify model parameters.

| | **FT (LLM)** | | | |
|---|---|---|---|---|
| **Model** | **Max Steps** | **Edit Layer** | **Optimizer** | **Edit LR** |
| DeSTA2.5-Audio | 15 | 31$^{st}$ layer of Transformer Module | Adam | 1e-5 |
| Qwen2-Audio | 15 | 31$^{st}$ layer of Transformer Module | Adam | 1e-4 |
| | **FT (Audio)** | | | |
| **Model** | **Max Steps** | **Edit Layer** | **Optimizer** | **Edit LR** |
| DeSTA2.5-Audio | 15 | perception.connector | Adam | 1e-4 |
| Qwen2-Audio | 15 | multi_modal_projector | Adam | 1e-4 |
| | **KE** | | | |
| **Model** | **Epoch** | **Edit Layer** | **Optimizer** | **LR** |
| DeSTA2.5-Audio | 1 | layer 29. 30, 31 of Transformer Module | RMSprop | 3e-4 |
| Qwen2-Audio | 1 | layer 29. 30, 31 of Transformer Module | RMSprop | 3e-4 |
| | **MEND** | | | |
| **Model** | **Epoch** | **Edit Layer** | **Optimizer** | **LR** |
| DeSTA2.5-Audio | 1 | layer 29. 30, 31 of Transformer Module | Adam | 1e-6 |
| Qwen2-Audio | 1 | layer 29. 30, 31 of Transformer Module | Adam | 1e-6 |
| | **UnKE** | | | |
| **Model** | **v_step/optim_step** | **Edit Layer** | **preserve_data** | **v_LR/optim_LR** |
| DeSTA2.5-Audio | 25/50 | layer 15 of Transformer Module | 20(text)/5(audio) | 5e-1/2e-4 |
| Qwen2-Audio | 25/50 | layer 20 of Transformer Module | 20(text)/5(audio) | 5e-1/2e-4 |
| | **WISE** | | | |
| **Model** | **$\alpha$, $\beta$, $\gamma$, act_ratio** | **Edit Layer** | **preserve_data** | **edit_LR/n_iter** |
| DeSTA2.5-Audio | 2,20,10,0.88 | layer 29 of Transformer Module | 10(text)/10(audio) | 0.1/50 |
| Qwen2-Audio | 5,20,10,0.88 | layer 26 of Transformer Module | 10(text)/10(audio) | 1.0/50 |

Table 5: Approximate execution time of each editing method on an NVIDIA H100 GPU, measured for training the trainer, single editing, and sequential editing.

| **Method** | **Model** | **Execution Time** | | |
|---|---|---|---|---|
| | | **Training Trainer** | **Single Editing** | **Sequential Editing** |
| **FT (LLM)** | DeSTA2.5-Audio | - | 5h 50m | 2h 0m |
| | Qwen2-Audio | - | 2h 35m | 1h 15m |
| **FT (Audio)** | DeSTA2.5-Audio | - | 4h 20m | 2h 0m |
| | Qwen2-Audio | - | 3h 0m | 1h 30m |
| **KE** | DeSTA2.5-Audio | 4h 10m | 8h 40m | 9h 45m |
| | Qwen2-Audio | 2h 10m | 4h 50m | 3h 20m |
| **MEND** | DeSTA2.5-Audio | 4h 10m | 4h 0m | 4h 10m |
| | Qwen2-Audio | 2h 20m | 2h 10m | 1h 50m |
| **UnKE** | DeSTA2.5-Audio | - | 3h 0m | 1h 05m |
| | Qwen2-Audio | - | 1h 35m | 0h 30m |
| **I-IKE** | DeSTA2.5-Audio | - | 6h 0m | 3h 45m |
| | Qwen2-Audio | - | 3h 0m | 2h 0m |
| **IE-IKE** | DeSTA2.5-Audio | - | 3h 20m | 8h 0m |
| | Qwen2-Audio | - | 2h 0m | 3h 50m |
| **WISE** | DeSTA2.5-Audio | - | 6h 10m | 2h 0m |
| | Qwen2-Audio | - | 3h 0m | 1h 10m |

**LLM Judge Prompt for Correctness**

**Task.** You will be given a question, a list of possible options, a ground-truth answer, and a model-generated response. Determine whether the model-generated response is *correct* according to the criteria below.

**Criteria.**

1. There is one and only one correct answer. Judge *incorrect* if the response does not choose any option from the list, or if it chooses more than one option.

2. If exactly one option from the list is chosen, judge *correct* if the chosen option aligns with the ground-truth answer; otherwise, judge *incorrect*.

3. Read the question, options, ground-truth answer, and model-generated response carefully before deciding.

**Examples.**

**Example 1.** *Question:* What is the capital of France?
   *Options:* Paris; London; Berlin; Madrid.
   *Ground truth answer:* Paris
   *Response:* "The capital of France is Tokyo." ⇒ *incorrect* (does not choose any listed option).
   *Response:* "The capital of France is Paris and London." ⇒ *incorrect* (chooses more than one option).
   *Response:* "The capital of France is London." ⇒ *incorrect* (chosen option does not match ground truth).
   *Response:* "The capital of France is Paris." ⇒ *correct*.

**Example 2.** *Question:* What is the underlying emotion of the speaker?
   *Options:* Happy; Sad; Angry; Neutral.
   *Ground truth answer:* Happy
   *Response:* "The speaker is happy." ⇒ *correct*.
   *Response:* "The speaker expresses happiness." ⇒ *correct* ("happiness" is a valid morphological variant of "happy").
   *Response:* "Happiness." ⇒ *correct*.

**Now judge:**
*Question:* {question}
*Ground truth answer:* {ground_truth}
*Model-generated response:* {model_response}

**Return format.**
```
Explanation: <Your explanation on your judgement>
Judgement: <correct or incorrect>
```

---

## LLM Judge Prompt for Consistency

**Role.** You are an expert evaluator tasked with determining whether two model outputs are *semantically consistent* for the same input.

**Inputs.**
*Question:* {question}
*Original Output:* {original_output}
*New Output:* {new_output}

**Task.** Decide if these two outputs are semantically consistent.

- Outputs are *consistent* if they express the same main message, even with different wording, phrasing, or level of detail.

- Outputs are *inconsistent* if they contradict each other, reach different conclusions, or provide fundamentally different answers.

**Return format.**
```
Explanation: <Your explanation on your judgement>
Judgement: <consistent or inconsistent>
```

Table 6: Detailed results of the four metrics of each auditory attribute across different editing methods on DeSTA2.5-Audio under single editing. Attr. denotes auditory attributes, and Port. denotes portability. For generality and audio locality, Avg. indicates the average performance across all types of the corresponding metric. (%)

| Method | Attr. | Reliability | Generality | | | | Audio Locality | | | | | Text Locality | Port. |
|---|---|---|---|---|---|---|---|---|---|---|---|---|---|
| | | | Avg. | Type 1 | Type 2 | Type 3 | Avg. | Type 1 | Type 2 | Type 3 | Type 4 | | |
| **FT (LLM)** | ALL | 99.75 | 98.75 | 99.67 | 99.00 | 97.58 | 65.11 | 88.08 | 15.56 | 74.67 | 69.75 | 82.58 | 19.42 |
| | Animal | 99.33 | 99.11 | 99.67 | 99.33 | 98.33 | 55.08 | 90.67 | 9.00 | 56.33 | 64.33 | 82.00 | 20.00 |
| | Emotion | 100.00 | 99.56 | 99.33 | 100.00 | 99.33 | 53.25 | 81.67 | 7.00 | 59.67 | 64.67 | 79.33 | 23.00 |
| | Gender | 99.67 | 96.78 | 99.67 | 96.67 | 94.00 | 84.67 | 90.33 | - | 88.33 | 75.33 | 86.33 | 7.00 |
| | Language | 100.00 | 99.56 | 100.00 | 100.00 | 98.67 | 72.33 | 89.67 | 30.67 | 94.33 | 74.67 | 82.67 | 27.67 |
| **FT (Audio)** | ALL | 99.50 | 86.14 | 96.75 | 84.92 | 76.75 | 68.09 | 78.67 | 48.11 | 76.17 | 64.42 | 100.00 | 55.50 |
| | Animal | 99.00 | 85.78 | 97.33 | 83.00 | 77.00 | 62.00 | 81.67 | 40.67 | 60.00 | 65.67 | 100.00 | 30.33 |
| | Emotion | 99.67 | 78.67 | 95.33 | 73.67 | 67.00 | 60.92 | 77.67 | 38.67 | 66.67 | 60.67 | 100.00 | 46.33 |
| | Gender | 99.67 | 92.33 | 99.33 | 92.67 | 85.00 | 77.67 | 78.33 | - | 87.33 | 67.33 | 100.00 | 82.33 |
| | Language | 99.67 | 87.78 | 95.00 | 90.33 | 78.00 | 74.17 | 77.00 | 65.00 | 90.67 | 64.00 | 100.00 | 63.00 |
| **KE** | ALL | 99.58 | 99.17 | 99.25 | 99.25 | 99.00 | 79.29 | 96.33 | 43.89 | 76.33 | 91.75 | 92.50 | 18.42 |
| | Animal | 98.67 | 97.33 | 97.67 | 97.33 | 97.00 | 77.33 | 97.00 | 61.67 | 59.00 | 91.67 | 91.33 | 23.67 |
| | Emotion | 100.00 | 100.00 | 100.00 | 100.00 | 100.00 | 66.17 | 98.33 | 16.33 | 59.33 | 90.67 | 93.33 | 16.00 |
| | Gender | 99.67 | 99.56 | 99.33 | 99.67 | 99.67 | 93.89 | 96.33 | - | 92.33 | 93.00 | 92.67 | 9.33 |
| | Language | 100.00 | 99.78 | 100.00 | 100.00 | 99.33 | 83.42 | 93.67 | 53.67 | 94.67 | 91.67 | 92.67 | 24.67 |
| **MEND** | ALL | 95.33 | 95.00 | 95.83 | 95.17 | 94.00 | 71.47 | 93.42 | 17.22 | 74.67 | 87.00 | 92.08 | 19.42 |
| | Animal | 90.00 | 89.44 | 91.33 | 89.00 | 88.00 | 63.50 | 93.67 | 19.33 | 57.00 | 84.00 | 92.00 | 20.33 |
| | Emotion | 99.67 | 98.44 | 99.00 | 99.00 | 97.33 | 62.25 | 94.33 | 9.00 | 59.67 | 86.00 | 94.00 | 27.33 |
| | Gender | 98.67 | 97.00 | 97.33 | 99.33 | 94.33 | 91.89 | 97.33 | - | 88.33 | 90.00 | 90.00 | 4.33 |
| | Language | 93.00 | 95.11 | 95.67 | 93.33 | 96.33 | 73.33 | 88.33 | 23.33 | 93.67 | 88.00 | 92.33 | 25.67 |
| **UnKE** | ALL | 96.33 | 89.28 | 89.67 | 94.25 | 83.92 | 56.04 | 67.67 | 10.56 | 71.92 | 62.67 | 87.42 | 18.75 |
| | Animal | 99.33 | 95.67 | 98.00 | 98.33 | 90.67 | 51.83 | 69.33 | 15.67 | 59.67 | 62.67 | 86.00 | 27.33 |
| | Emotion | 96.00 | 91.56 | 91.33 | 94.33 | 89.00 | 50.42 | 66.33 | 12.00 | 57.00 | 66.33 | 87.67 | 20.00 |
| | Gender | 99.67 | 86.11 | 85.67 | 99.67 | 73.00 | 72.33 | 58.67 | - | 94.33 | 64.00 | 86.67 | 7.33 |
| | Language | 90.33 | 83.78 | 83.67 | 84.67 | 83.00 | 53.67 | 76.33 | 4.00 | 76.67 | 57.67 | 89.33 | 20.33 |
| **I-IKE** | ALL | 73.00 | 61.47 | 64.67 | 59.58 | 60.17 | 65.40 | 79.42 | 50.67 | 73.67 | 54.17 | 62.08 | 71.42 |
| | Animal | 90.33 | 92.89 | 92.67 | 93.00 | 93.00 | 54.75 | 84.33 | 22.00 | 59.67 | 53.00 | 60.00 | 78.33 |
| | Emotion | 59.33 | 50.89 | 58.00 | 47.00 | 47.67 | 60.08 | 80.67 | 45.33 | 62.33 | 52.00 | 59.00 | 65.33 |
| | Gender | 69.33 | 64.33 | 71.33 | 60.33 | 61.33 | 73.33 | 81.67 | - | 79.67 | 58.67 | 64.33 | 73.00 |
| | Language | 38.67 | 37.78 | 36.67 | 38.00 | 38.67 | 75.42 | 71.00 | 84.67 | 93.00 | 53.00 | 65.00 | 69.00 |
| **IE-IKE** | ALL | 40.58 | 39.61 | 41.50 | 38.67 | 38.67 | 58.76 | 70.08 | 49.89 | 70.75 | 42.08 | 56.25 | 34.58 |
| | Animal | 77.00 | 78.56 | 80.00 | 76.67 | 79.00 | 51.17 | 78.67 | 23.33 | 59.00 | 43.67 | 60.33 | 44.33 |
| | Emotion | 27.00 | 26.67 | 28.67 | 25.33 | 26.00 | 51.33 | 68.00 | 33.33 | 61.33 | 42.67 | 51.33 | 29.00 |
| | Gender | 41.67 | 37.67 | 43.00 | 36.00 | 34.00 | 61.67 | 74.67 | - | 67.67 | 42.67 | 60.33 | 32.67 |
| | Language | 16.67 | 15.56 | 14.33 | 16.67 | 15.67 | 71.58 | 59.00 | 93.00 | 95.00 | 39.33 | 53.00 | 32.33 |
| **WISE** | ALL | 100.00 | 100.00 | 100.00 | 100.00 | 100.00 | 37.00 | 27.85 | 3.22 | 76.00 | 26.62 | 70.25 | 11.40 |
| | Animal | 100.00 | 100.00 | 100.00 | 100.00 | 100.00 | 31.67 | 35.67 | 3.33 | 58.00 | 29.77 | 71.00 | 7.00 |
| | Emotion | 100.00 | 99.89 | 100.00 | 100.00 | 99.67 | 23.25 | 15.67 | 6.33 | 56.33 | 14.67 | 61.33 | 0.60 |
| | Gender | 100.00 | 100.00 | 100.00 | 100.00 | 100.00 | 60.77 | 42.33 | - | 93.67 | 46.33 | 78.00 | 27.67 |
| | Language | 100.00 | 100.00 | 100.00 | 100.00 | 100.00 | 32.33 | 17.73 | 0.00 | 96.00 | 15.72 | 70.67 | 10.33 |

## F    DETAILED RESULTS FOR EACH AUDITORY ATTRIBUTE UNDER SINGLE EDITING

We compare the performance for each auditory attribute (**animal** sound, speaker **emotion**, speaker **gender**, and spoken **language**) under single editing here, as shown in Table 6 and Table 7 for DeSTA2.5-Audio and Qwen2-Audio, respectively.

**Reliability.** All methods except I-IKE and IE-IKE exhibit consistently high reliability across all attributes, suggesting that they can effectively update the edited knowledge on both LALMs. In contrast, the performance of I-IKE and IE-IKE varies by attribute and model, which we attribute to differences in the original model's ability to perceive auditory attributes.

**Generality.** For most methods, their generality remains relatively stable across different auditory attributes. However, FT (Audio) exhibits a notable exception: although it achieves consistently high

Table 7: Detailed results of the four metrics of each auditory attribute across different editing methods on Qwen2-Audio under single editing. Attr. denotes auditory attributes, and Port. denotes portability. For generality and audio locality, Avg. indicates the average performance across all types of the corresponding metric. (%)

| Method | Attr. | Relability | Generality | | | | Audio Locality | | | | | Text Locality | Port. |
|---|---|---|---|---|---|---|---|---|---|---|---|---|---|
| | | | Avg. | Type 1 | Type 2 | Type 3 | Avg. | Type 1 | Type 2 | Type 3 | Type 4 | | |
| **FT (LLM)** | ALL | 100.00 | 99.94 | 100.00 | 100.00 | 99.83 | 67.42 | 91.67 | 10.44 | 83.33 | 70.00 | 74.58 | 24.67 |
| | Animal | 100.00 | 99.89 | 100.00 | 100.00 | 99.67 | 64.00 | 93.00 | 8.67 | 86.33 | 68.00 | 70.00 | 22.67 |
| | Emotion | 100.00 | 100.00 | 100.00 | 100.00 | 100.00 | 59.00 | 93.67 | 8.00 | 66.00 | 68.33 | 80.00 | 20.67 |
| | Gender | 100.00 | 100.00 | 100.00 | 100.00 | 100.00 | 83.78 | 85.67 | - | 92.33 | 73.33 | 77.67 | 26.33 |
| | Language | 100.00 | 99.89 | 100.00 | 100.00 | 99.67 | 67.00 | 94.33 | 14.67 | 88.67 | 70.33 | 70.67 | 29.00 |
| **FT (Audio)** | ALL | 100.00 | 81.86 | 99.00 | 77.17 | 69.42 | 90.53 | 96.83 | 80.11 | 90.00 | 92.58 | 100.00 | 50.67 |
| | Animal | 100.00 | 84.44 | 99.00 | 81.00 | 73.33 | 94.25 | 98.00 | 94.00 | 93.33 | 91.67 | 100.00 | 48.67 |
| | Emotion | 100.00 | 71.67 | 99.00 | 62.67 | 53.33 | 85.08 | 98.33 | 67.00 | 84.33 | 90.67 | 100.00 | 43.00 |
| | Gender | 100.00 | 97.44 | 100.00 | 96.00 | 96.33 | 94.56 | 98.00 | - | 92.00 | 93.67 | 100.00 | 63.00 |
| | Language | 100.00 | 73.89 | 98.00 | 69.00 | 54.67 | 89.25 | 93.00 | 79.33 | 90.33 | 94.33 | 100.00 | 48.00 |
| **KE** | ALL | 95.50 | 86.67 | 92.00 | 87.67 | 80.33 | 83.47 | 89.83 | 61.44 | 87.25 | 89.83 | 84.92 | 27.58 |
| | Animal | 97.00 | 88.22 | 89.33 | 94.00 | 81.33 | 88.42 | 93.67 | 76.33 | 92.33 | 91.33 | 81.33 | 24.67 |
| | Emotion | 98.33 | 76.67 | 91.33 | 74.33 | 64.33 | 78.00 | 97.33 | 52.67 | 72.00 | 90.00 | 87.00 | 23.33 |
| | Gender | 87.67 | 89.22 | 90.33 | 88.33 | 89.00 | 84.78 | 72.67 | - | 93.67 | 88.00 | 85.00 | 39.00 |
| | Language | 99.00 | 92.56 | 97.00 | 94.00 | 86.67 | 83.00 | 95.67 | 55.33 | 91.00 | 90.00 | 86.33 | 23.33 |
| **MEND** | ALL | 100.00 | 95.33 | 98.92 | 95.92 | 91.17 | 83.27 | 98.50 | 49.33 | 85.42 | 91.33 | 86.75 | 27.17 |
| | Animal | 100.00 | 97.33 | 98.00 | 98.33 | 95.67 | 84.50 | 97.00 | 64.00 | 89.67 | 87.33 | 83.67 | 22.67 |
| | Emotion | 100.00 | 89.89 | 99.00 | 92.00 | 78.67 | 75.67 | 100.00 | 42.00 | 70.33 | 90.33 | 89.00 | 23.67 |
| | Gender | 100.00 | 95.89 | 100.00 | 94.33 | 93.33 | 94.89 | 99.00 | - | 92.33 | 93.33 | 88.67 | 37.00 |
| | Language | 100.00 | 98.22 | 98.67 | 99.00 | 97.00 | 80.92 | 98.00 | 42.00 | 89.33 | 94.33 | 85.67 | 25.33 |
| **UnKE** | ALL | 98.58 | 98.53 | 99.08 | 98.83 | 97.67 | 67.49 | 91.42 | 12.34 | 82.92 | 69.50 | 71.58 | 28.42 |
| | Animal | 98.00 | 97.67 | 97.67 | 96.33 | 99.00 | 63.50 | 88.00 | 10.67 | 86.67 | 68.67 | 73.33 | 22.33 |
| | Emotion | 96.67 | 97.33 | 99.00 | 99.67 | 93.33 | 62.25 | 95.33 | 18.67 | 64.67 | 70.33 | 70.33 | 23.00 |
| | Gender | 100.00 | 99.56 | 100.00 | 99.67 | 99.00 | 82.67 | 90.00 | - | 91.67 | 66.33 | 72.00 | 39.33 |
| | Language | 99.67 | 99.56 | 99.67 | 99.67 | 99.33 | 65.33 | 92.33 | 7.67 | 88.67 | 72.67 | 70.67 | 29.00 |
| **I-IKE** | ALL | 10.33 | 7.11 | 10.33 | 5.67 | 5.33 | 87.51 | 94.75 | 89.33 | 93.00 | 73.42 | 55.00 | 28.92 |
| | Animal | 6.67 | 7.11 | 7.00 | 7.33 | 7.00 | 90.08 | 96.33 | 94.00 | 94.33 | 75.67 | 54.33 | 25.00 |
| | Emotion | 19.67 | 13.22 | 19.67 | 9.00 | 11.00 | 82.58 | 97.00 | 77.67 | 83.33 | 72.33 | 58.67 | 23.00 |
| | Gender | 9.67 | 3.11 | 8.67 | 0.67 | 0.00 | 88.22 | 94.00 | - | 99.00 | 71.67 | 56.00 | 44.00 |
| | Language | 5.33 | 5.00 | 6.00 | 5.67 | 3.33 | 89.33 | 91.67 | 96.33 | 95.33 | 74.00 | 51.00 | 23.67 |
| **IE-IKE** | ALL | 8.00 | 6.58 | 8.50 | 5.75 | 5.50 | 82.89 | 91.25 | 86.56 | 89.67 | 65.00 | 50.00 | 27.50 |
| | Animal | 6.33 | 6.67 | 7.33 | 6.00 | 6.67 | 84.17 | 90.33 | 90.67 | 88.33 | 67.33 | 51.33 | 26.00 |
| | Emotion | 9.67 | 11.67 | 11.67 | 11.00 | 12.33 | 79.50 | 96.67 | 76.33 | 80.33 | 64.67 | 51.00 | 21.67 |
| | Gender | 11.00 | 3.78 | 10.67 | 0.67 | 0.00 | 83.67 | 89.00 | - | 99.00 | 63.00 | 50.67 | 40.33 |
| | Language | 5.00 | 4.22 | 4.33 | 5.33 | 3.00 | 84.42 | 89.00 | 92.67 | 91.00 | 65.00 | 47.00 | 22.00 |
| **WISE** | ALL | 100.00 | 99.31 | 99.83 | 99.17 | 98.92 | 69.02 | 92.42 | 5.67 | 82.67 | 74.67 | 64.50 | 27.00 |
| | Animal | 100.00 | 99.56 | 99.83 | 99.67 | 99.67 | 63.25 | 90.67 | 2.67 | 86.00 | 73.67 | 63.66 | 6.00 |
| | Emotion | 100.00 | 98.78 | 100.00 | 98.67 | 97.67 | 59.83 | 92.67 | 9.67 | 65.00 | 72.00 | 62.66 | 23.00 |
| | Gender | 100.00 | 98.89 | 100.00 | 98.33 | 98.33 | 87.56 | 94.00 | - | 91.00 | 77.00 | 70.00 | 42.00 |
| | Language | 100.00 | 100.00 | 100.00 | 100.00 | 100.00 | 65.42 | 92.33 | 4.67 | 88.67 | 76.00 | 61.67 | 37.00 |

reliability, its generality on emotion attributes drops the most compared to its reliability on both LALMs. This suggests that editing the modality connector makes it harder for the model to extend edited knowledge to similar inputs within emotion.

**Locality.** Most methods achieve relatively higher average performance on the gender attribute, owing to the inapplicability of type 2 audio locality, which was identified in Sec. 5.1 as the most difficult to preserve, thereby inflating the overall average. For attributes other than gender, most methods perform comparably across attributes on Qwen2-Audio. In contrast, on DeSTA2.5-Audio, although performance on type 1 and type 4 is generally similar, higher audio locality is observed for language in type 2 and type 3. This indicates that unrelated auditory knowledge regarding spoken language is less susceptible to disruption when edits are applied on DeSTA2.5-Audio. Regarding text locality, most methods demonstrate comparable performance across attributes on both LALMs.

**Portability.** Different methods result in varying portability performance across attributes on the two LALMs. A consistent observation on both models is that FT (Audio) achieves higher portability scores for all atributes, especially gender, suggesting that editing the modality connector may more effectively propagate the edited knowledge to other interconnected knowledge, particularly that related to speaker gender.

# G    DETAILED RESULTS UNDER SEQUENTIAL EDITING

We provide the statistics of the sequential editing results in Table 8 and 9, with the corresponding line charts shown in Figure 3a and Figure 3b for DeSTA2.5-Audio and Qwen2-Audio, respectively.

Table 8: Original result of the four metrics of different editing methods on DeSTA2.5-Audio under sequential editing. For generality and audio locality, we present the averaged results. (%)

| Method | Gap | Reliability | Generality | Audio Locality | Text Locality | Portability |
|--------|-----|-------------|------------|----------------|---------------|-------------|
| FT (LLM) | 0 | 100.00 | 99.33 | 49.74 | 82.00 | 8.00 |
| | 1 | 78.00 | 75.33 | 48.19 | 70.00 | 8.00 |
| | 2 | 68.00 | 68.67 | 46.11 | 74.00 | 10.00 |
| | 3 | 64.00 | 57.33 | 45.08 | 72.00 | 10.00 |
| | 4 | 58.00 | 53.33 | 41.45 | 74.00 | 12.00 |
| | 5 | 54.00 | 48.00 | 45.60 | 72.00 | 12.00 |
| FT (Audio) | 0 | 98.00 | 88.00 | 53.89 | 100.00 | 52.00 |
| | 1 | 78.00 | 62.00 | 49.74 | 100.00 | 38.00 |
| | 2 | 64.00 | 46.00 | 41.97 | 100.00 | 30.00 |
| | 3 | 62.00 | 48.67 | 41.45 | 100.00 | 24.00 |
| | 4 | 52.00 | 38.00 | 39.38 | 100.00 | 40.00 |
| | 5 | 46.00 | 36.00 | 40.93 | 100.00 | 32.00 |
| MEND | 0 | 22.00 | 18.67 | 26.94 | 54.00 | 4.00 |
| | 1 | 12.00 | 9.33 | 13.47 | 28.00 | 8.00 |
| | 2 | 4.00 | 4.67 | 2.59 | 14.00 | 2.00 |
| | 3 | 2.00 | 2.00 | 0.52 | 4.00 | 2.00 |
| | 4 | 2.00 | 2.00 | 1.04 | 6.00 | 0.00 |
| | 5 | 0.00 | 0.00 | 0.00 | 2.00 | 0.00 |
| KE | 0 | 62.00 | 64.00 | 40.41 | 76.00 | 16.00 |
| | 1 | 36.00 | 36.00 | 34.20 | 58.00 | 14.00 |
| | 2 | 12.00 | 14.67 | 29.02 | 56.00 | 4.00 |
| | 3 | 6.00 | 16.67 | 24.35 | 38.00 | 8.00 |
| | 4 | 26.00 | 16.00 | 22.28 | 28.00 | 10.00 |
| | 5 | 8.00 | 8.67 | 16.58 | 40.00 | 4.00 |
| UnKE | 0 | 92.00 | 89.33 | 45.08 | 82.00 | 34.00 |
| | 1 | 52.00 | 34.67 | 38.34 | 88.00 | 18.00 |
| | 2 | 38.00 | 26.67 | 32.64 | 86.00 | 12.00 |
| | 3 | 22.00 | 20.07 | 37.82 | 82.00 | 30.00 |
| | 4 | 26.00 | 23.33 | 37.82 | 84.00 | 24.00 |
| | 5 | 22.00 | 10.67 | 33.68 | 80.00 | 24.00 |
| IE-IKE | 0 | 32.00 | 30.00 | 58.55 | 54.00 | 28.00 |
| | 1 | 26.00 | 36.67 | 59.07 | 58.00 | 34.00 |
| | 2 | 30.00 | 35.33 | 58.03 | 50.00 | 32.00 |
| | 3 | 32.00 | 34.00 | 54.92 | 54.00 | 34.00 |
| | 4 | 32.00 | 39.33 | 53.37 | 54.00 | 38.00 |
| | 5 | 26.00 | 32.00 | 52.33 | 54.00 | 34.00 |
| I-IKE | 0 | 60.00 | 54.67 | 65.28 | 50.00 | 56.00 |
| | 1 | 56.00 | 50.67 | 60.10 | 52.00 | 52.00 |
| | 2 | 56.00 | 54.67 | 61.66 | 68.00 | 54.00 |
| | 3 | 56.00 | 52.00 | 62.69 | 52.00 | 48.00 |
| | 4 | 62.00 | 56.67 | 61.14 | 52.00 | 46.00 |
| | 5 | 62.00 | 56.67 | 59.07 | 56.00 | 54.00 |
| WISE | 0 | 100.00 | 100.00 | 39.90 | 74.00 | 22.00 |
| | 1 | 78.00 | 75.33 | 39.90 | 80.00 | 14.00 |
| | 2 | 76.00 | 68.00 | 36.27 | 78.00 | 20.00 |
| | 3 | 74.00 | 66.00 | 34.72 | 76.00 | 16.00 |
| | 4 | 70.00 | 61.33 | 36.79 | 78.00 | 16.00 |
| | 5 | 62.00 | 54.66 | 36.27 | 74.00 | 16.00 |

Table 9: Original result of the four metrics of different editing methods on Qwen2-Audio under sequential editing. For generality and audio locality, we present the averaged results. (%)

| Method | Gap | Reliability | Generality | Audio Locality | Text Locality | Portability |
|---|---|---|---|---|---|---|
| FT (LLM) | 0 | 100.00 | 100.00 | 44.56 | 64.00 | 18.00 |
| | 1 | 82.00 | 74.67 | 41.97 | 70.00 | 16.00 |
| | 2 | 72.00 | 61.33 | 41.97 | 64.00 | 18.00 |
| | 3 | 70.00 | 55.33 | 42.49 | 64.00 | 14.00 |
| | 4 | 66.00 | 51.33 | 45.08 | 62.00 | 20.00 |
| | 5 | 56.00 | 43.33 | 40.41 | 66.00 | 16.00 |
| FT (Audio) | 0 | 100.00 | 82.67 | 74.09 | 100.00 | 42.00 |
| | 1 | 100.00 | 79.33 | 72.54 | 100.00 | 40.00 |
| | 2 | 100.00 | 77.33 | 71.50 | 100.00 | 40.00 |
| | 3 | 100.00 | 76.00 | 70.98 | 100.00 | 36.00 |
| | 4 | 100.00 | 76.00 | 70.98 | 100.00 | 34.00 |
| | 5 | 100.00 | 75.33 | 66.84 | 100.00 | 40.00 |
| MEND | 0 | 88.00 | 88.67 | 64.25 | 86.00 | 20.00 |
| | 1 | 76.00 | 72.67 | 68.91 | 82.00 | 20.00 |
| | 2 | 66.00 | 68.67 | 65.28 | 66.00 | 16.00 |
| | 3 | 68.00 | 68.00 | 64.25 | 72.00 | 22.00 |
| | 4 | 64.00 | 66.67 | 58.55 | 70.00 | 14.00 |
| | 5 | 64.00 | 55.33 | 57.51 | 70.00 | 18.00 |
| KE | 0 | 72.00 | 70.67 | 41.97 | 54.00 | 12.00 |
| | 1 | 34.00 | 27.33 | 29.02 | 40.00 | 8.00 |
| | 2 | 22.00 | 17.33 | 22.28 | 32.00 | 2.00 |
| | 3 | 10.00 | 17.33 | 18.65 | 24.00 | 4.00 |
| | 4 | 8.00 | 10.67 | 13.47 | 14.00 | 2.00 |
| | 5 | 14.00 | 9.33 | 13.47 | 12.00 | 4.00 |
| UnKE | 0 | 100.00 | 98.00 | 47.67 | 66.00 | 26.00 |
| | 1 | 54.00 | 54.00 | 37.82 | 62.00 | 10.00 |
| | 2 | 50.00 | 50.00 | 38.86 | 54.00 | 24.00 |
| | 3 | 42.00 | 34.67 | 40.93 | 50.00 | 22.00 |
| | 4 | 38.00 | 29.33 | 29.53 | 52.00 | 24.00 |
| | 5 | 40.00 | 26.00 | 33.16 | 36.00 | 16.00 |
| IE-IKE | 0 | 10.00 | 6.00 | 77.20 | 38.00 | 26.00 |
| | 1 | 14.00 | 7.33 | 72.02 | 40.00 | 22.00 |
| | 2 | 4.00 | 6.67 | 55.44 | 34.00 | 26.00 |
| | 3 | 14.00 | 6.00 | 42.49 | 34.00 | 20.00 |
| | 4 | 8.00 | 6.67 | 27.46 | 20.00 | 10.00 |
| | 5 | 4.00 | 4.00 | 20.73 | 14.00 | 8.00 |
| I-IKE | 0 | 10.00 | 7.33 | 76.17 | 48.00 | 26.00 |
| | 1 | 10.00 | 7.33 | 77.20 | 44.00 | 22.00 |
| | 2 | 10.00 | 8.00 | 75.13 | 48.00 | 26.00 |
| | 3 | 10.00 | 8.00 | 75.65 | 52.00 | 20.00 |
| | 4 | 10.00 | 8.67 | 75.65 | 42.00 | 26.00 |
| | 5 | 10.00 | 6.67 | 77.72 | 44.00 | 22.00 |
| WISE | 0 | 100.00 | 100.00 | 58.55 | 60.00 | 14.00 |
| | 1 | 92.00 | 90.00 | 57.51 | 58.00 | 10.00 |
| | 2 | 92.00 | 78.67 | 62.18 | 60.00 | 14.00 |
| | 3 | 98.00 | 82.67 | 60.62 | 58.00 | 14.00 |
| | 4 | 94.00 | 80.67 | 61.66 | 58.00 | 20.00 |
| | 5 | 96.00 | 80.00 | 63.21 | 50.00 | 18.00 |

## H  MORE DISCUSSIONS OF INTRA-ATTRIBUTE KNOWLEDGE ENTANGLEMENT

In Sec. 5.1, we found that type 2 audio locality is hardest to preserve. We hypothesize that the severe failure in Audio Locality Type 2 arises because intra-attribute knowledge often has much blurrier boundaries than inter-attribute knowledge, and thus is more susceptible to both acoustic and semantic entanglement.

**Acoustic Entanglement** Unlike inter-attribute distinctions (e.g., emotions versus animal sounds), which are naturally disparate, labels within the same attribute share substantial acoustic characteristics. FFor example, within the emotion attribute, both "sad" and "fearful" voices may sound quieter or more tense, and both often have slower speaking rates. These prosodic and spectral similarities imply that editing one label cen easily perturbs the representations of closely related labels, making intra-attribute knowledge difficult to preserve.

**Semantic Entanglement** As shown in Table 1, our experimental results indicate that FT(LLM) performs significantly worse than FT(Audio) on Audio Locality Type 2. This disparity suggests that intra-attribute knowledge is more easily entangled when the LLM backbone is modified. A probable explanation is that intra-attribute distinctions rely on fine-grained semantic cues that occupy narrow margins in the model's hidden space. When editing these parameters directly, the update may cause the model to conflate closely related labels.

In summary, current editing methods do not explicitly account for these subtle intra-attribute relationships, suggesting that both acoustic and semantic factors jointly contribute to the observed failures. Developing methods that can better probe, analyze, and preserve intra-attribute boundaries is a valuable direction for future work.

## I  DETAILED BREAKDOWN OF PORTABILITY EVALUATION

Given that the scope of potential related knowledge is vast and cannot be entirely enumerated, SAKE addresses this by aggregating portability evaluations across a diverse set of editing instances. Specifically, within our dataset, multiple editing instances may share the same editing pair $(y_o, y_e)$; however, they are assessed using different portability questions that target a wide variety of related concepts of the edited attribute. To provide a granular analysis of knowledge portability across these concepts, we detail the specific categories covered for each auditory attribute and present the performance breakdown under single editing.

### I.1  TAXONOMY OF REASONING CONCEPTS

The portability questions are categorized based on the following related concepts:

**Animal Sounds**

- **Behavior:** Typical behavior of the animal (e.g., purring, chewing cud).
- **Care Item:** Required items for caring for the animal (e.g., litter box, leash).
- **Diet:** Dietary classification of the animal.
- **Family:** The closest animal in terms of biological taxonomy.
- **Locomotion:** The way the animal moves.
- **Physical:** Physical traits of the animal (e.g., claws, hooves).
- **Reproduction:** The reproductive method of the animal.
- **Vocalization:** The characteristic vocalization of the animal.

**Speaker Emotion**

- **Descriptive Sentence:** A sentence reflecting the speaker's emotional state.
- **Facial Expression:** The facial expression representing the emotion.
- **Scenario:** The scenario or situation that matches the speaker's emotion.
- **Social Interaction:** The appropriate social interaction as a response based on the emotion.

**Speaker Gender**

- **History:** A historical figure who shares the same gender as the speaker.

- **Celebrity:** A celebrity who shares the same gender as the speaker.
- **Cloth:** Traditional clothing associated with the speaker's gender in the relevant cultural context.
- **Title:** Formal title of the speaker (e.g., Mr., Ms.).
- **Vocal:** Vocal ranges that align with the speaker's gender.

**Spoken Language**

- **ISO:** The ISO language code of the spoken language.
- **City:** A city in a country where the spoken language is recognized as official.
- **Dish:** A dish originating from a country where the language is official.
- **History:** A historical figure from a country where the language is official.
- **Language Family:** The language family to which the language belongs.
- **Literary:** Books or literary works originally authored in the language.
- **Official Language:** The country that recognizes the spoken language as official.
- **Test:** A test used to evaluate proficiency in the language.
- **Translation:** The translation of an English word into the spoken language.

## I.2    PERFORMANCE BREAKDOWN

Tables 10, 11, 12, and 13 detail the portability breakdown under single editting for *Animal Sounds*, *Speaker Emotion*, *Speaker Gender*, and *Spoken Language* attributes, respectively. The analysis reveals a mixed state of reasoning propagation. Rather than a binary outcome where all related concepts succeed or fail simultaneously, performance varies by category. For instance, within the *Animal Sound* attribute, FT(Audio) shows a disparity in knowledge propagation, achieving 45.95% accuracy on "reproduction" compared to only 7.89% on "family." Overall, instances where related concepts fail to update still constitute the majority, indicating that current editing methods struggle to consistently propagate knowledge across different reasoning dimensions.

Table 10: Breakdown of portability performance across different concept categories for the *Animal Sound* attribute. (%)

| Model | Method | Behavior | Care Item | Diet | Family | Locomotion | Physical | Reproduction | Vocalization |
|---|---|---|---|---|---|---|---|---|---|
| DeSTA2.5-Audio | FT (Audio) | 2.70 | 23.68 | 21.05 | 2.63 | 68.57 | 13.51 | 16.22 | 15.00 |
| | FT (Audio) | 27.03 | 39.47 | 23.68 | 7.89 | 25.71 | 32.43 | 45.95 | 40.00 |
| | KE | 10.81 | 42.11 | 5.26 | 26.32 | 34.29 | 21.62 | 10.81 | 37.50 |
| | MEND | 5.41 | 42.11 | 39.47 | 10.53 | 28.57 | 10.81 | 18.92 | 7.50 |
| | UnKE | 24.32 | 52.63 | 26.32 | 7.89 | 31.43 | 10.81 | 18.92 | 45.00 |
| | I-IKE | 83.78 | 92.11 | 68.42 | 47.37 | 85.71 | 78.38 | 78.38 | 92.50 |
| | IE-IKE | 35.14 | 57.89 | 42.11 | 13.16 | 51.43 | 37.84 | 35.14 | 80.00 |
| | WISE | 0.00 | 2.63 | 0.00 | 0.00 | 0.00 | 0.00 | 0.00 | 2.50 |
| Qwen2-Audio | FT (LLM) | 24.32 | 26.32 | 15.79 | 23.68 | 31.43 | 21.62 | 8.11 | 30.00 |
| | FT (Audio) | 59.46 | 71.05 | 36.84 | 26.32 | 37.14 | 51.35 | 45.95 | 60.00 |
| | KE | 27.03 | 36.84 | 15.79 | 23.68 | 31.43 | 13.51 | 8.11 | 40.00 |
| | MEND | 24.32 | 39.47 | 18.42 | 15.79 | 11.43 | 16.22 | 21.62 | 32.50 |
| | UnKE | 45.95 | 21.05 | 21.05 | 13.16 | 0.00 | 10.81 | 13.51 | 50.00 |
| | I-IKE | 32.43 | 36.84 | 21.05 | 18.42 | 34.29 | 10.81 | 5.41 | 40.00 |
| | IE-IKE | 27.03 | 50.00 | 21.05 | 21.05 | 22.86 | 16.22 | 5.41 | 42.50 |
| | WISE | 18.92 | 13.16 | 0.00 | 2.63 | 0.00 | 0.00 | 0.00 | 12.50 |

Table 11: Breakdown of portability performance across different concept categories for the *Speaker Emotion* attribute. (%)

| Model | Method | Descriptive Sentence | Facial Expression | Scenario | Social Interaction |
|---|---|---|---|---|---|
| DeSTA2.5-Audio | FT (Audio) | 24.32 | 22.08 | 18.42 | 27.40 |
| | FT (Audio) | 44.59 | 48.05 | 47.37 | 45.21 |
| | KE | 17.57 | 14.29 | 11.84 | 20.55 |
| | MEND | 24.32 | 27.27 | 28.95 | 28.77 |
| | UnKE | 18.92 | 23.38 | 17.11 | 20.55 |
| | I-IKE | 71.62 | 51.95 | 68.42 | 69.86 |
| | IE-IKE | 47.30 | 23.38 | 19.74 | 26.03 |
| | WISE | 8.11 | 5.19 | 15.79 | 4.11 |
| Qwen2-Audio | FT (Audio) | 21.62 | 15.58 | 13.16 | 32.88 |
| | FT (Audio) | 47.30 | 44.16 | 35.53 | 45.21 |
| | KE | 22.97 | 15.58 | 22.37 | 32.88 |
| | MEND | 22.97 | 15.58 | 23.68 | 32.88 |
| | UnKE | 22.97 | 15.58 | 22.37 | 31.51 |
| | I-IKE | 22.97 | 16.88 | 17.11 | 35.62 |
| | IE-IKE | 21.62 | 18.18 | 18.42 | 28.77 |
| | WISE | 29.73 | 7.79 | 22.37 | 32.88 |

Table 12: Breakdown of portability performance across different concept categories for the *Speaker Gender* attribute. (%)

| Model | Method | History | Celebrity | Cloth | Title | Vocal |
|---|---|---|---|---|---|---|
| DeSTA2.5-Audio | FT (Audio) | 6.35 | 7.46 | 3.33 | 6.56 | 12.24 |
| | FT (Audio) | 85.71 | 79.10 | 70.00 | 93.44 | 83.67 |
| | KE | 9.52 | 4.48 | 16.67 | 4.92 | 12.24 |
| | MEND | 4.76 | 2.99 | 3.33 | 3.28 | 8.16 |
| | UnKE | 4.76 | 7.46 | 11.67 | 4.92 | 8.16 |
| | I-IKE | 82.54 | 53.73 | 70.00 | 78.69 | 83.67 |
| | IE-IKE | 36.51 | 23.88 | 18.33 | 45.90 | 40.82 |
| | WISE | 1.59 | 1.49 | 3.33 | 3.28 | 0.00 |
| Qwen2-Audio | FT (Audio) | 49.21 | 11.94 | 18.33 | 32.79 | 18.37 |
| | FT (Audio) | 90.48 | 37.31 | 40.00 | 90.16 | 57.14 |
| | KE | 60.32 | 31.34 | 36.67 | 40.98 | 22.45 |
| | MEND | 61.90 | 19.40 | 28.33 | 52.46 | 20.41 |
| | UnKE | 58.73 | 35.82 | 36.67 | 36.07 | 26.53 |
| | I-IKE | 77.78 | 26.87 | 28.33 | 55.74 | 28.57 |
| | IE-IKE | 65.08 | 28.36 | 35.00 | 36.07 | 36.73 |
| | WISE | 69.84 | 23.88 | 28.33 | 60.66 | 24.49 |

Table 13: Breakdown of portability performance across different concept categories for the *Spoken Language* attribute. (%)

| Model | Method | Iso | City | Dish | History | Language Family | Literary | Official Language | Test | Translation |
|---|---|---|---|---|---|---|---|---|---|---|
| DeSTA2.5-Audio | FT (Audio) | 10.71 | 44.12 | 37.14 | 36.11 | 14.81 | 2.86 | 26.47 | 30.77 | 40.62 |
| | FT (Audio) | 67.86 | 58.82 | 74.29 | 66.67 | 74.07 | 54.29 | 58.82 | 38.46 | 81.25 |
| | KE | 17.86 | 26.47 | 22.86 | 36.11 | 14.81 | 17.14 | 26.47 | 15.38 | 43.75 |
| | MEND | 14.29 | 26.47 | 28.57 | 30.56 | 11.11 | 17.14 | 5.88 | 43.59 | 46.88 |
| | UnKE | 32.14 | 26.47 | 14.29 | 16.67 | 18.52 | 14.29 | 8.82 | 20.51 | 34.38 |
| | I-IKE | 82.14 | 70.59 | 77.14 | 72.22 | 62.96 | 48.57 | 61.76 | 64.10 | 84.38 |
| | IE-IKE | 57.14 | 26.47 | 37.14 | 22.22 | 11.11 | 28.57 | 23.53 | 23.08 | 65.62 |
| | WISE | 0.00 | 0.00 | 0.00 | 0.00 | 0.00 | 5.71 | 2.94 | 0.00 | 0.00 |
| Qwen2-Audio | FT (Audio) | 3.57 | 52.94 | 34.29 | 38.89 | 3.70 | 5.71 | 26.47 | 33.33 | 53.12 |
| | FT (Audio) | 75.00 | 44.12 | 45.71 | 38.89 | 44.44 | 34.29 | 67.65 | 28.21 | 62.50 |
| | KE | 3.57 | 29.41 | 28.57 | 19.44 | 14.81 | 20.00 | 11.76 | 30.77 | 46.88 |
| | MEND | 17.86 | 32.35 | 25.71 | 13.89 | 18.52 | 20.00 | 26.47 | 25.64 | 46.88 |
| | UnKE | 57.14 | 29.41 | 17.14 | 19.44 | 40.74 | 11.43 | 11.76 | 17.95 | 68.75 |
| | I-IKE | 10.71 | 35.29 | 20.00 | 19.44 | 18.52 | 14.29 | 20.59 | 23.08 | 50.00 |
| | IE-IKE | 7.14 | 26.47 | 25.71 | 13.89 | 29.63 | 14.29 | 5.88 | 25.64 | 50.00 |
| | WISE | 28.57 | 44.12 | 42.86 | 38.89 | 18.52 | 8.57 | 44.12 | 43.59 | 59.38 |

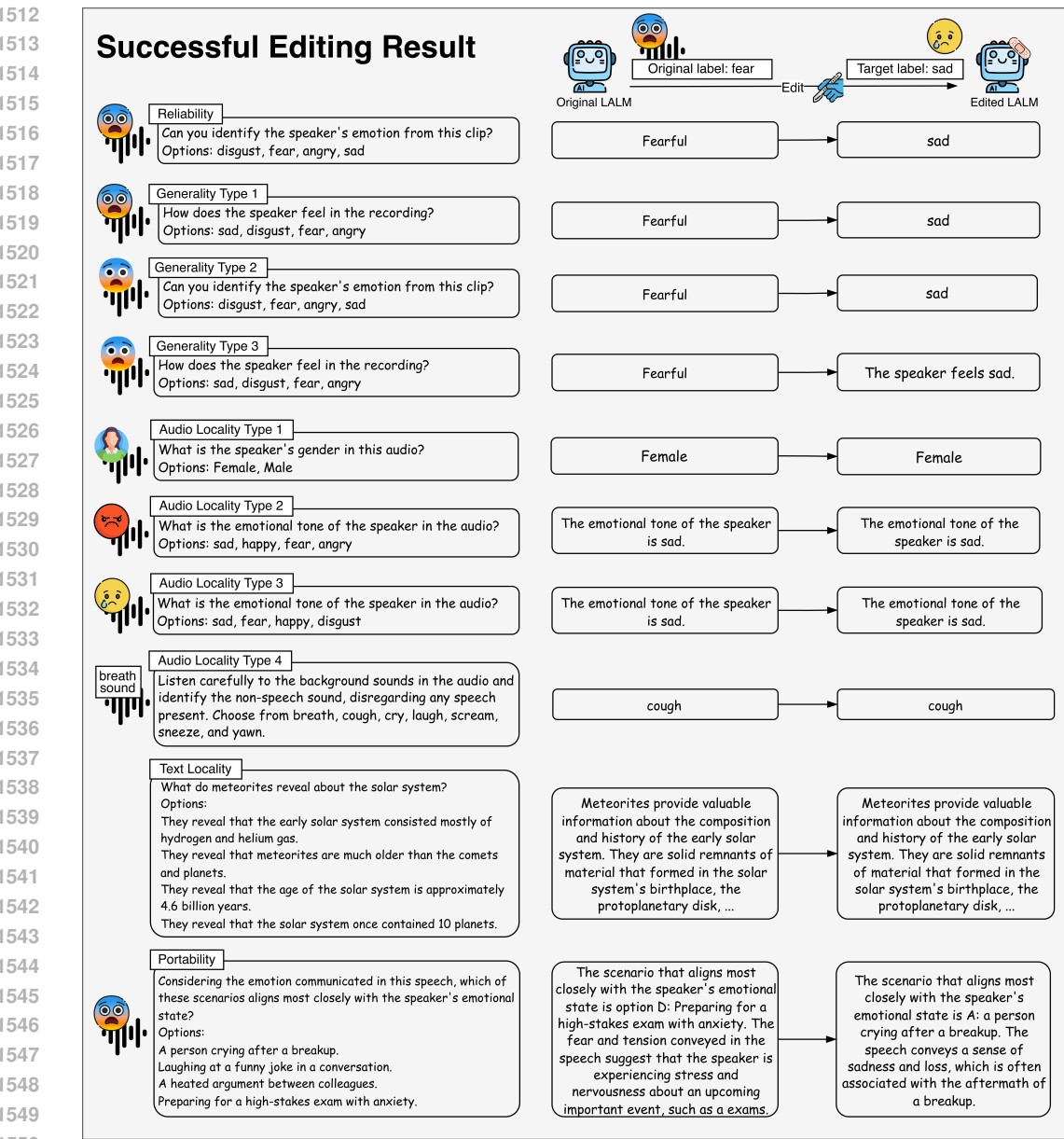

Figure 4: An example of a successful editing result by FT (Audio) on Qwen2-Audio.

## J CASE STUDY

While knowledge editing in LALMs offers a way to update auditory knowledge without full retraining, the editing process is not always stable. To better understand the behaviors, we conduct a case study comparing two editing scenarios: (1) a single targeted edit applied to change an emotional attribute, and (2) sequential edits applied across multiple concepts. The comparison highlights both the strengths and limitations of current editing methods, emphasizing the trade-off between reliability of isolated edits and the accumulation of errors when multiple edits interact.

Figure 4 shows the result of a successful editing example by FT (Audio) on Qwen2-Audio, where we edit the model's perception of speaker emotion from "fearful" to "sad." After editing, we observe that the model's outputs for both reliability and generality are successfully updated to "sad." At the same time, both audio locality and text locality remain intact, which shows that the original knowledge of the model is preserved. For portability, we observe that after the edit, the model

can also perform reasoning, changing the prediction from "preparing for a high-stakes exam with anxiety" to "a person crying after a breakup." This demonstrates that the interconnected knowledge is also updated during reasoning, and the edited model can apply this knowledge in new contexts.

Figure 5 shows a degenerated example from sequential editing on DeSTA2.5-Audio with MEND. After applying multiple edits in a row, we can see that the model collapses and produces incoherent outputs such as repeated characters and newline symbols. This illustrates how sequential edits can accumulate interference and destabilize the internal representations of the model. Unlike the single-edit scenario, where the change is targeted and localized, multiple edits interact in unpredictable ways, leading to corruption of reliability, loss of generality, and failure of both locality and portability. To be practical in real-world scenarios, however, an editing method must be capable of supporting many edits simultaneously while ensuring that unrelated edits do not interfere with one another. This failure case thus underscores a key limitation of current approaches.

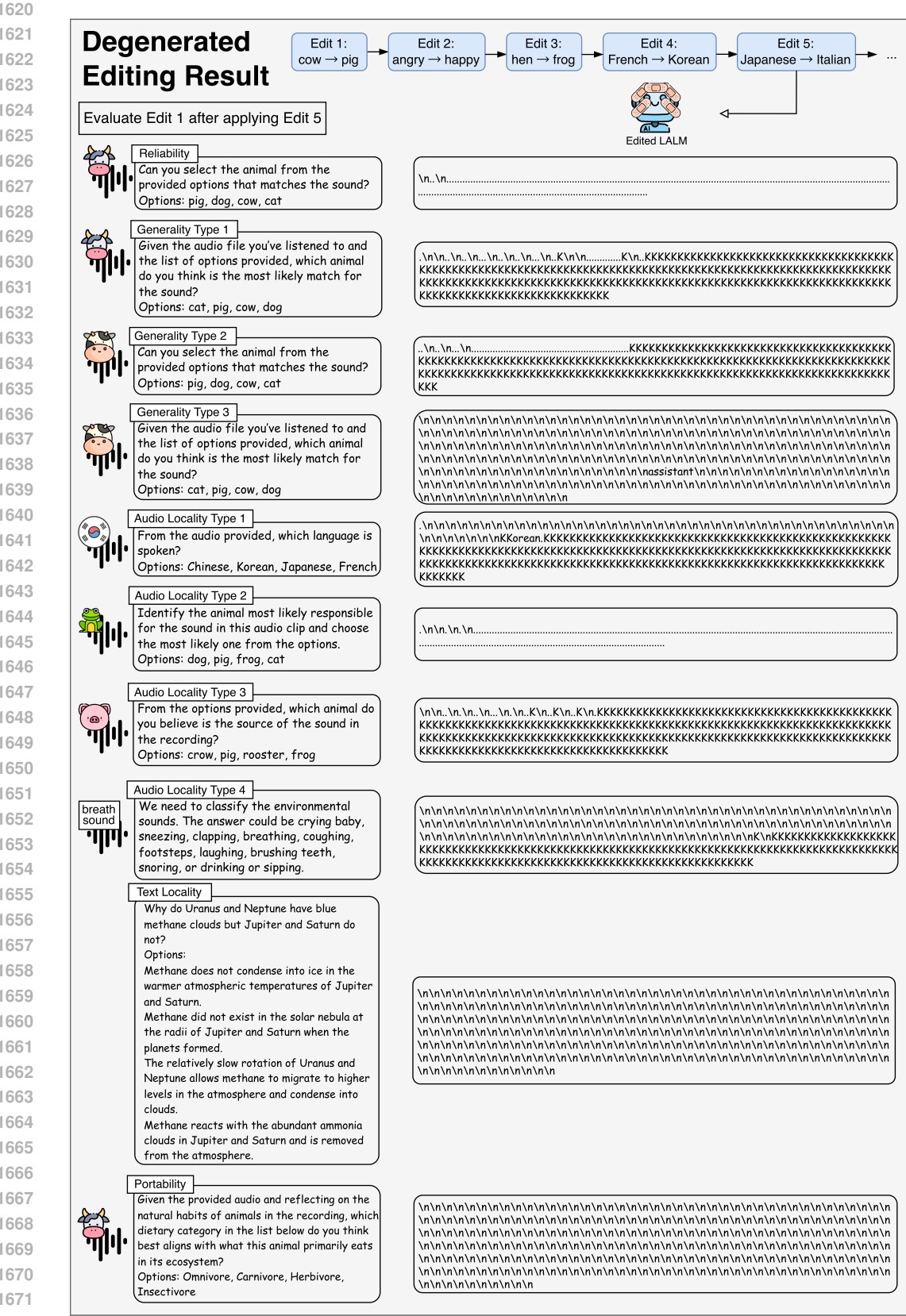

Figure 5: An example of a degenerated editing result by MEND on DeSTA2.5-Audio.

