# OpenReview forum: "SAKE: Towards Editing Auditory Attribute Knowledge of Large Audio-Language Models"
_ICLR.cc/2026/Conference — Submitted to ICLR 2026_

### Official Review · Reviewer_c6Zy · 2025-11-01

**Soundness:** 3
**Presentation:** 3
**Contribution:** 3
**Rating:** 6
**Confidence:** 4

**Summary:**

This paper introduces SAKE, the first benchmark specifically designed to evaluate knowledge editing in Large Audio-Language Models (LALMs). The authors posit that editing auditory attribute knowledge (e.g., speaker emotion, animal sounds) presents unique challenges not found in text or vision, as these attributes are abstract, high-level perceptual concepts rather than discrete facts.

**Strengths:**

* **Pioneering Problem Definition:** This is the first work to formally define and systematically benchmark the problem of editing auditory attribute knowledge in LALMs, moving beyond traditional text/vision fact editing.
* **Rigorous Benchmark Framework:** The SAKE benchmark is methodologically strong. Its design, particularly the granular "Audio Locality Type 2" (intra-attribute) and "Portability" (reasoning propagation) metrics, astutely captures the unique, abstract challenges of the auditory modality.
* **Significant and Actionable Findings:** The paper delivers clear, impactful results. The discovery of "intra-attribute entanglement" (e.g., editing "dog" breaks "cat") and the documented failure of IKE methods for LALMs are crucial findings that highlight specific, unsolved challenges for the field.
* **High-Quality Presentation:** The paper is exceptionally clear. Figure 1, in particular, serves as an excellent visual abstract that effectively communicates the benchmark's complex design and evaluation dimensions.

**Weaknesses:**

* **Mismatch in Attribute Scope:** The paper's motivation hinges on editing "abstract and continuous" auditory concepts, yet the benchmark's attributes (e.g., animal sound, language) are predominantly evaluated as discrete classification labels. This under-delivers on the core premise, as the challenges of editing truly continuous attributes (like pitch or prosody) remain unexplored.
* **Superficial Analysis of IKE Failure:** The paper reports the stark failure of In-Context Editing (IKE) but attributes it to a generic "limited in-context learning ability." This analysis is shallow; it fails to investigate the specific failure mechanism, such as whether the LALM struggles to process in-context audio or fails to apply textual instructions to its auditory processing.
* **Limited Model Diversity:** Key findings, such as "intra-attribute entanglement," are derived from only two LALMs. While acknowledged as a limitation, this narrow scope makes it difficult to ascertain if these significant challenges are fundamental to LALMs or are artifacts of the specific architectures tested.

**Questions:**

1.  **IKE Failure Mechanism:** Given the failure of In-Context Editing, is this due to the LALM's inability to process in-context *audio examples*, or a more general failure to apply *textual instructions* to its auditory processing? A test using only text-based instructions could isolate the precise point of failure.
2.  **Intra-Attribute Entanglement Mechanism:** What is the hypothesized cause for the severe "Audio Locality Type 2" failure (e.g., editing "dog" breaks "cat")? Is it (a) **acoustic entanglement**, where representations are similar in the audio encoder, or (b) **semantic entanglement**, where the LLM backbone co-locates these concepts?
3.  **Scope of Portability Failure:** The portability test (animal sound $\rightarrow$ diet) fails. How wide is this reasoning disconnect? For the "frog" $\rightarrow$ "dog" edit, do other related concepts like "habitat" (pond $\rightarrow$ house) or "classification" (amphibian $\rightarrow$ mammal) also fail to update, or is the failure isolated to the tested attribute?

---

> ### Author Response · Authors · 2025-11-17
> **Response to Weakness 2 and Question 1**
>
> We believe the reviewer may have overlooked some details regarding our analysis of IKE. In our experiments, we evaluate two variants of in-context editing: Instruction-based IKE (I-IKE), which encodes the edit exclusively through natural language instructions in the system prompt, and Instruction+Example IKE (IE-IKE), which supplements instructions with in-context auditory examples. The former directly corresponds to the scenario the reviewer described, namely applying textual instructions to guide the model’s auditory processing. Although I-IKE performs better than IE-IKE, its overall performance remains unsatisfactory.
>
> These results indicate that the failure of IKE is not limited to processing in-context audio but also extends to processing purely textual instructions for editing auditory knowledge. Since both forms of in-context conditioning fail to reliably induce the desired edits, we summarize this observation as limited in-context learning ability in both textual and auditory modalities.
>
> We believe this comment arises from a misunderstanding of our work, and we hope this clarification addresses the reviewer’s concern.

---

> ### Author Response · Authors · 2025-11-20
> **Response to Weakness 3**
>
> We selected two LALMs that are widely recognized and among the state-of-the-art models available at the time of writing. Importantly, the focus of our work is to evaluate the performance of editing methods rather than the underlying models themselves. Our evaluation examines whether an edit is successful, which includes producing the desired outputs when the edited knowledge is required and preserving knowledge that is unrelated to the edit. Since our primary goal is to compare editing techniques, it is both reasonable and standard practice to select a small number of representative models as the test bed and apply a broad range of editing methods on top of them. Using two strong and widely adopted LALMs, therefore, aligns with established methodology in the knowledge editing literature, as noted in our limitations section. We appreciate the reviewer’s suggestion, but we believe this should not be regarded as a weakness of the work.
>
> Furthermore, DeSTA2.5-Audio and Qwen2-Audio are developed using notably different modeling philosophies, such as LLM backbones (LLaMA vs. Qwen) and modality adapter designs (using Q-former or not), as well as training pipelines (e.g., using self-generated data or not). Given these substantial differences, it is unlikely that the observed challenges, such as the intra-attribute entanglement, are artifacts unique to these two specific LALMs. Instead, these findings consistently emerge across multiple editing methods and across both models, suggesting that the difficulties stem from limitations of existing editing techniques themselves. We therefore believe the phenomena we report reflect more fundamental challenges in auditory knowledge editing.
>
> We hope this clarification addresses the reviewer’s concern.

---

> ### Author Response · Authors · 2025-11-24
> **Response to Question 2**
>
> We thank the reviewer for this insightful question. We hypothesize that the severe failure in Audio Locality Type 2 arises because intra-attribute knowledge often has much blurrier boundaries than inter-attribute knowledge, and thus is more susceptible to **both acoustic and semantic entanglement.**
>
> - **Acoustic entanglement**: Different attributes, such as emotions versus animal sounds, are naturally distinct and therefore easier to separate. In contrast, labels within the same attribute frequently share many common characteristics and differ only in subtle ways. For example, within the emotion attribute, both “sad” and “fearful” voices may sound quieter or more tense, and both often have slower speaking rates. Their differences, such as specific prosodic contours or slight changes in vocal tension, are subtler, making their acoustic characteristics more similar. Because these labels are closely related, editing one label can more easily influence another, making it harder to preserve intra-attribute knowledge.
> - **Semantic entanglement**: Our experimental results (Table 1) show that FT(LLM) performs much worse than FT(Audio) on Audio Locality Type 2. This suggests that intra-attribute knowledge is more easily entangled when the LLM backbone is modified. A possible explanation is that intra-attribute distinctions also rely on fine-grained semantic cues that lie on narrow margins in the model’s hidden space. When editing these parameters directly, the update may cause the model to conflate closely related labels.
>
> Given these observations, it is difficult to isolate a single cause for intra-attribute entanglement. We hypothesize that both acoustic and semantic factors jointly contribute to the failure observed in Audio Locality Type 2.
>
> We appreciate that the reviewer recognizes our findings on intra-attribute entanglement as a meaningful contribution of the paper. This insight is enabled by our benchmark design, which evaluates knowledge locality at a finer granularity than existing multimodal knowledge-editing benchmarks in the vision domain. Prior benchmarks often rely on locality samples that are obviously unrelated to the edited knowledge, which prevents the discovery of such subtle interactions. In contrast, SAKE introduces multiple locality types, including the explicit evaluation of intra-attribute knowledge, which enables us to reveal behaviors that had not been previously observed. As the first benchmark for knowledge editing in LALMs, SAKE provides the foundation needed for a deeper understanding of how auditory attribute knowledge behaves under editing.
>
> We agree that further investigating the mechanism underlying intra-attribute entanglement is an important direction. Our empirical results suggest that existing editing methods do not explicitly account for these subtle intra-attribute relationships, which may explain their limited performance. Developing methods that can better probe, analyze, and preserve intra-attribute boundaries is a valuable direction for future work, and we plan to explore it in subsequent research.

---

> ### Author Response · Authors · 2025-11-24
> **Response to Weakness 1**
>
> We thank the reviewer for raising this concern. We would like to clarify that in our paper, the term “continuous” does not refer to the labels themselves being continuous. Instead, it describes the perceptual and acoustic nature of the underlying auditory attributes, which exhibit rich variability and do not have clear or discrete boundaries. Each attribute encompasses a wide range of realizations with subtle but meaningful differences.
>
> For example, animal sounds are continuous in nature because vocalizations from the same species can vary substantially across individuals. Speaker emotion is also continuous because different speakers express the same emotion differently, and even a single emotion can vary in intensity, prosody, or timbre. Spoken language exhibits continuous variation across speakers due to accent, articulation patterns, and speaking style. Even speaker gender, though often categorized discretely, reflects continuous differences in acoustic features such as pitch range, formant configuration, and vocal resonance.
>
> In contrast, textual factual knowledge is typically represented through explicit triplets of subjects, relations, and objects, which are far more concrete and discretely structured. Auditory attributes, by comparison, are perceptual concepts that cannot be exhaustively enumerated or represented.
>
> For this reason, although our evaluation uses discrete labels as anchors, the underlying challenge remains the same: the model must update or preserve boundaries within a continuous and high-variance acoustic distribution that inherently involves pitch, prosody, timbre, and other auditory cues. The discrete labels are simply a practical mechanism for verifying whether the underlying continuous perceptual concepts have been edited correctly.
>
> We hope this clarification resolves the reviewer’s concern, and we will refine the wording in the revised manuscript to prevent any similar misunderstandings in the future.

---

> ### Author Response · Authors · 2025-12-02
> **Response to Question 3**
>
> We appreciate the reviewer’s thoughtful question. Although the scope of potential related concepts is vast and cannot be entirely enumerated, our benchmark addresses this by aggregating portability evaluations across a diverse set of editing instances ($A \to B$). Specifically, for a given edit type (e.g., Frog $\to$ Dog), our dataset includes multiple editing instances. Each instance is paired with a different portability question targeting a wide variety of reasoning dimensions (e.g., one instance tests "diet," while another tests "behavior"). Since most methods exhibit high reliability (near 100%, as shown in Table 1 in our paper) except for two IKE variants, this design allows us to statistically estimate the portability performance across various concepts. The specific concept categories included for each auditory attribute and the performance breakdown have been provided in Appendix I of our updated paper. Please refer to it for more details.
>
> Our granular analysis reveals that the reasoning disconnect is wide but not uniform. **It is not a binary outcome where all concepts succeed or fail simultaneously; rather, we see a mixed state.** For example, within the animal sound attribute, the result of the FT(Audio) method suggests that the model might better propagate knowledge regarding “reproduction” (45.95%) compared to “family” (7.89%) after editing. Nevertheless, our results indicate that currently, instances where related concepts fail to update constitute the majority. This underscores the need for further investigation into the causes of these failures and the development of new editing methods in future work.

---

### Official Review · Reviewer_gqjC · 2025-11-01

**Soundness:** 3
**Presentation:** 3
**Contribution:** 3
**Rating:** 4
**Confidence:** 3

**Summary:**

This paper introduces SAKE, the first benchmark designed to evaluate knowledge editing in Large Audio-Language Models. SAKE targets auditory attribute knowledge, such as speaker gender, emotion, spoken language, and animal sounds.
The benchmark evaluates seven editing methods across four dimensions: Reliability, Generality, Locality, Portability
Experiments were conducted on two strong LALMs: DeSTA2.5-Audio and Qwen2-Audio.
The results show that while existing editing methods can successfully change specific auditory knowledge, they struggle to generalize, maintain unrelated knowledge, and support multiple sequential edits.
The paper concludes that new methods are needed to handle abstract, perceptual auditory knowledge more robustly

**Strengths:**

- First benchmark focused on auditory attribute knowledge editing, extending a well-studied concept from text and vision into the audio domain.
- Significant for maintaining and updating multimodal model knowledge efficiently.
- Results are well-analyzed, identifying causes of poor generality and locality in existing methods.

**Weaknesses:**

- Only 2 LALMs are evaluated.
- Only 4 attributes are covered. Can other auditory attributes like environmental sound types, etc. be considered for a stronger benchmark?

**Questions:**

- Are the paraphrased text human-checked for semantic consistency?
- Can the authors show evaluation on recent LALMs like Audio Flamingo 2, etc.?

---

> ### Author Response · Authors · 2025-11-17
> **Response to the Weakness 1**
>
> We selected two LALMs that are widely recognized and among the state-of-the-art models available at the time of writing. Importantly, the focus of our work is to evaluate the performance of editing methods, not the underlying models themselves. Our evaluation examines whether an edit is successful, which includes producing the desired outputs when the edited knowledge is required and preserving knowledge that is unrelated to the edit. Since the primary goal is to compare editing techniques, it is both reasonable and standard practice to select some representative models as the test bed and apply a broad set of editing methods on top of them.
>
> Hence, using two strong and widely adopted LALMs as the evaluation base aligns with common methodology in the knowledge editing literature, as we also state in our limitations. We appreciate the reviewer’s suggestion, but we believe this aspect should not be regarded as a weakness of the work.

---

> ### Author Response · Authors · 2025-11-17
> **Response to Weakness 2**
>
> We selected these four attributes for two main reasons. First, they are fundamental to speech and audio processing. Competence in these attributes underpins a variety of real-world applications, such as emotion recognition and affective dialogue systems. They are also widely covered in existing LALM benchmarks [1-5], underscoring their central role in the field. As this work represents the first attempt to investigate knowledge editing for auditory modalities, we focus on these core attributes as an initial and representative starting point.
>
> Second, because of their importance, there exist comparatively rich resources for these attributes, including both audio datasets and question–answer pairs. This is particularly relevant for our portability track, which requires multi-hop reasoning [4] rather than simple attribute recognition. The availability of sufficiently large and diverse data makes these attributes more suitable for constructing a reliable, scalable, and comprehensive benchmark.
>
> Finally, we acknowledge that incorporating additional auditory attributes in the future would further broaden the scope and utility of the benchmark. However, we believe that the current selection does not represent a deficiency in the work. Rather, it provides a focused and principled foundation for the first benchmark on auditory knowledge editing, upon which broader extensions can naturally be built.
>
>
> [1] Huang et al., "Dynamic-SUPERB: Towards A Dynamic, Collaborative, and Comprehensive Instruction-Tuning Benchmark for Speech", ICASSP 2024
>
> [2] Huang et al., "Dynamic-SUPERB Phase-2: A Collaboratively Expanding Benchmark for Measuring the Capabilities of Spoken Language Models with 180 Tasks", ICLR 2025
>
> [3] Yang et al., "AIR-Bench: Benchmarking Large Audio-Language Models via Generative Comprehension", ACL 2024
>
> [4] Yang et al., "SAKURA: On the Multi-hop Reasoning of Large Audio-Language Models Based on Speech and Audio Information", Interspeech 2025
>
> [5] Sakshi et al., "MMAU: A Massive Multi-Task Audio Understanding and Reasoning Benchmark", ICLR 2025

---

> ### Author Response · Authors · 2025-11-17
> **Responses to Questions**
>
> # Question 1
>
> Yes, all the paraphrased data are manually verified. We will mention this in the updated version of the paper. Thank you for pointing this out.
>
> # Question 2
>
> In our previous responses, we clarified the general rationale for using two LALMs. Here, we further explain why we specifically selected Qwen2-Audio and DeSTA2.5-Audio.
>
> Both Qwen2-Audio and DeSTA2.5-Audio are widely recognized and among the state-of-the-art open-source LALMs available at the time of writing. We include Qwen2-Audio because it is one of the most broadly adopted models in LALM research, including studies on reasoning [1, 2], safety [3, 4], interpretability [5], and broader multimodal evaluation [6]. It therefore serves as a highly representative model within the LALM community.
>
> DeSTA2.5-Audio, on the other hand, is one of the most recent models at the time of submission, and in fact is newer than the Audio Flamingo 2 model suggested by the reviewer. It also achieves leading performance on several benchmarks, such as SAKURA [1] and MMAU [2], which makes it an appropriate representative of recent advancements in LALMs.
>
> These considerations justify our choice of models. We would also like to emphasize that the primary focus of this work is the evaluation of knowledge editing methods rather than the comparison of LALMs themselves. As a result, it is standard practice to select a small number of representative models as the test bed on which different editing methods are evaluated. We hope this explanation addresses the reviewer’s concern.
>
> [1] Yang et al., "SAKURA: On the Multi-hop Reasoning of Large Audio-Language Models Based on Speech and Audio Information", Interspeech 2025
>
> [2] Sakshi et al., "MMAU: A Massive Multi-Task Audio Understanding and Reasoning Benchmark", ICLR 2025
>
> [3] Yang et al., "Audio Is the Achilles’ Heel: Red Teaming Audio Large Multimodal Models", NAACL 2025
>
> [4] Roh et al., "Multilingual and Multi-Accent Jailbreaking of Audio LLMs", COLM 2025
>
> [5] Yang et al., "AudioLens: A Closer Look at Auditory Attribute Perception of Large Audio-Language Models", ASRU 2025
>
> [6] Huang et al., "Dynamic-SUPERB Phase-2: A Collaboratively Expanding Benchmark for Measuring the Capabilities of Spoken Language Models with 180 Tasks", ICLR 2025

---

### Official Review · Reviewer_vcdK · 2025-11-01

**Soundness:** 2
**Presentation:** 2
**Contribution:** 2
**Rating:** 2
**Confidence:** 3

**Summary:**

This paper introduces SAKE, the first comprehensive evaluation benchmark specifically designed for auditory attribute knowledge editing in Large Audio-Language Models.
The SAKE benchmark evaluates editing capabilities across four core auditory attributes (speaker gender, emotion, language, and animal sounds) along four key dimensions: reliability, generality, locality, and portability. The authors conducted experiments on two leading LALMs (DeSTA2.5-Audio and Qwen2-Audio), evaluating seven common editing methods, including fine-tuning, Knowledge Editor, MEND, and In-Context Knowledge Editing in both single and sequential editing settings.

**Strengths:**

1. The topic is very interesting and well-motivated.
2.  The SAKE benchmark is designed with four critical dimensions: reliability, generality, locality, and portability. The benchmark has a potential impact on the following studies.

**Weaknesses:**

1. The evaluated scope of knowledge editing methods is limited. While seven editing methods were evaluated, some SOTA editing methods can be considered, such as WISE, AlphaEdit, UltraEdit.
2. The paper primarily focuses on evaluating the effects of editing (what works/doesn't work, and where). However, there's a relative lack of deeper mechanistic explanations for why certain methods are effective or ineffective in the auditory modality. For instance, which parameters or layers are most crucial for auditory attribute knowledge editing? What changes occur in the internal representations of the model? How is knowledge of different auditory attributes encoded and interlinked within the model? Incorporating interpretability analyses (e.g., feature attribution, probing tasks) to delve into the internal workings of knowledge editing on LALMs would provide more profound insights.
3. The considered auditory attributes are limited to 4. This may be due to the speech modality itself, but a more solid analysis is needed.
4. The edited performance can be related to the audio generation performance of the original model (without any editing). More baselines can be useful for discussion.
5. The editing task is limited to audio understanding, while audio generation can be a  much more significant scenario.

**Questions:**

Please see Weakness

---

> ### Author Response · Authors · 2025-11-17
> **Response to Weakness 3**
>
> We selected these four attributes for two main reasons. First, they are fundamental to speech and audio processing. Competence in these attributes underpins a variety of real-world applications, such as emotion recognition and affective dialogue systems. They are also widely covered in existing LALM benchmarks [1-5], underscoring their central role in the field. As this work represents the first attempt to investigate knowledge editing for auditory modalities, we focus on these core attributes as an initial and representative starting point.
>
> Second, because of their importance, there exist comparatively rich resources for these attributes, including both audio datasets and question–answer pairs. This is particularly relevant for our portability track, which requires multi-hop reasoning [4] rather than simple attribute recognition. The availability of sufficiently large and diverse data makes these attributes more suitable for constructing a reliable, scalable, and comprehensive benchmark.
>
> Finally, we acknowledge that incorporating additional auditory attributes in the future would further broaden the scope and utility of the benchmark. However, we believe that the current selection does not represent a deficiency in the work. Rather, it provides a focused and principled foundation for the first benchmark on auditory knowledge editing, upon which broader extensions can naturally be built.
>
>
> [1] Huang et al., "Dynamic-SUPERB: Towards A Dynamic, Collaborative, and Comprehensive Instruction-Tuning Benchmark for Speech", ICASSP 2024
>
> [2] Huang et al., "Dynamic-SUPERB Phase-2: A Collaboratively Expanding Benchmark for Measuring the Capabilities of Spoken Language Models with 180 Tasks", ICLR 2025
>
> [3] Yang et al., "AIR-Bench: Benchmarking Large Audio-Language Models via Generative Comprehension", ACL 2024
>
> [4] Yang et al., "SAKURA: On the Multi-hop Reasoning of Large Audio-Language Models Based on Speech and Audio Information", Interspeech 2025
>
> [5] Sakshi et al., "MMAU: A Massive Multi-Task Audio Understanding and Reasoning Benchmark", ICLR 2025

---

> ### Author Response · Authors · 2025-11-17
> **Response to Weakness 4**
>
> We would like to clarify the reviewer’s concern. In our experiments, the LALMs under evaluation are text-generating models rather than audio-generating models. Consequently, the notion of “audio generation performance” does not apply to our setting, since the models do not synthesize audio in any part of the benchmark.
>
> Regarding the choice of models, we selected two LALMs that are widely recognized and among the state-of-the-art models available at the time of writing. Their original capabilities are therefore sufficiently robust to support a meaningful study of auditory knowledge editing. Importantly, the focus of our work is to evaluate the performance of editing methods, not the performance of the underlying models themselves. Our evaluation examines whether an edit is successful, which includes producing the desired outputs when the edited knowledge is required and preserving knowledge that is unrelated to the edit. Since the primary goal is to compare editing techniques, it is both reasonable and standard practice to select some representative models as the test bed and apply a broad set of editing methods on top of them.
>
> While additional models could always be included, using two strong and widely adopted LALMs as the evaluation base aligns with common methodology in the knowledge editing literature, as we also state in our limitations section. We appreciate the reviewer’s suggestion, but we believe this aspect should not be regarded as a weakness of the work.

---

> ### Author Response · Authors · 2025-11-17
> **Response to Weakness 5**
>
> We appreciate the reviewer’s comment and agree that audio generation represents an important application where knowledge editing could also play a valuable role. This is indeed an interesting future direction.
>
> However, our work specifically focuses on text-generating LALMs rather than audio-generating models. Since the models used in this work do not synthesize audio, extending our benchmark to audio generation falls outside the scope of the current study.
> We would also like to emphasize that audio understanding itself is a crucial capability for a wide range of real-world applications, including emotion recognition, spoken language understanding, speaker profiling, affective dialogue systems, and downstream multimodal reasoning. These tasks rely heavily on robust auditory attribute understanding, and their practical significance is not necessarily lower than that of audio generation. Therefore, in our opinion, the suggestion that audio generation is the “more significant” scenario is not entirely appropriate, as the importance of these two directions depends on the application context rather than a strict hierarchy.
>
> Nonetheless, we appreciate the reviewer’s insight and will explicitly mention in the limitations section that extending SAKE to cover audio generation models is a valuable direction for future work.

---

> ### Author Response · Authors · 2025-11-26
> **Response to Weakness 2**
>
> We thank the reviewer for the thoughtful suggestion. We agree that interpretability analysis for knowledge editing in LALMs would be valuable. However, interpretability for LALMs remains an open research problem in itself. Only a very limited number of works have attempted to analyze LALMs from an interpretability perspective [1,2]. This highlights that developing tools capable of examining both auditory and textual modalities remains a significant unsolved challenge. Establishing such methods would already constitute a substantial standalone research problem.
>
> Nevertheless, our experiments still provide informative insights. For example, the comparison between FT (LLM) and FT (Audio) suggests that tuning the modality adapter tends to preserve locality better and achieve stronger portability, implying that components associated with modality bridging may be particularly relevant for editing auditory attribute knowledge. In addition, our benchmark reveals the difficulty of preserving intra-attribute knowledge, an observation that offers meaningful insights into how auditory knowledge may be entangled within current models.
>
> As the first work to explore knowledge editing for auditory attributes in LALMs, our primary objective is to establish a rigorous benchmark and characterize the limitations and behaviors of existing editing methods. Introducing a comprehensive interpretability framework is beyond the scope of this paper, especially given the lack of established tools for the auditory modality. We agree that mechanistic analysis is an interesting and important direction, and we will include this as future work.
>
>
>
> [1] Yang et al., “AudioLens: A Closer Look at Auditory Attribute Perception of Large Audio-Language Models”, ASRU 2025
>
> [2] Glazer et al., “Beyond Transcription: Mechanistic Interpretability in ASR”, arXiv preprint

---

> ### Author Response · Authors · 2025-12-02
> **Response to Weakness 1**
>
> We thank the reviewer for the suggestion. While our paper already evaluates seven editing methods, which we believe provides a sufficiently comprehensive comparison, we have additionally incorporated one more method during the discussion period. Specifically, we include **WISE** as an additional baseline. We selected WISE because it is both well-recognized and straightforward to implement. In contrast, UltraEdit is concurrent with our work and thus not yet widely established, and implementing AlphaEdit requires substantial memory resources that exceed our current computational capacity. For these reasons, we chose WISE.
>
> The introduction of WISE and its performance on our benchmark have been added to the revised version of the paper (Sec. 4.1, 5.1, and 5.2). Overall, WISE performs well in terms of reliability and generality. However, it performs poorly in knowledge locality and in updating relevant knowledge in the portability evaluation. This indicates that, despite being among the state-of-the-art editing methods in text-based LLM studies, WISE remains limited when applied to auditory attribute knowledge. These findings further highlight the challenges and unique difficulties of this research direction.

---

### Official Review · Reviewer_J6YR · 2025-11-01

**Soundness:** 2
**Presentation:** 3
**Contribution:** 2
**Rating:** 4
**Confidence:** 4

**Summary:**

The paper introduces SAKE, a benchmark for editing auditory attribute knowledge in Large Audio-Language Models (LALMs). It targets four attributes, including speaker gender, speaker emotion, spoken language, and animal sounds, and evaluates seven editing methods on two LALMs.

**Strengths:**

1. It is the first systematic benchmark for auditory attribute editing.
2. It tests two competitive LALMs, multiple attributes, single vs. sequential editing, and a comparative suite of seven methods.

**Weaknesses:**

1. Though the contribution is positioned primarily as a benchmark, the sample of edits provided is quite narrow and under specified. For example, when editing an attribute (e.g., changing “sad” → “angry”), the paper does not clearly define whether the edit is restricted to a specific sound instance or intended to generalize across all instances of the “sad” attribute. Without this scope clarity, it is difficult to interpret whether the model is simply mapping the one training instance or truly generalizing the attribute change.

2. In the locality evaluation, the benchmark requires that unrelated knowledge (i.e., non-edited items) remains unaffected. However, some of the design choices undermine this. For instance, in Figure 4 the locality sample has the answer “sad,” which exactly matches the original (pre-edited) attribute. This raises a concern: if the “locality” example uses the same attribute value as the edited item, then a correct response could simply reflect propagation of the edit rather than demonstrating true preservation of unrelated knowledge. The benchmark therefore may not reliably distinguish between genuine locality preservation and unintentional overlap with the edited attribute.

**Questions:**

Please refer to the Weakness.

---

> ### Author Response · Authors · 2025-11-17
> **Response to the Weakness 1**
>
> We thank the reviewer's comment. The edit is performed on a single instance. However, our evaluation explicitly covers both (1) the edited instance and (2) other instances sharing the same attribute. These correspond to our reliability and generality metrics, respectively. Therefore, the two aspects raised by Reviewer J6YR are already included in our benchmark. We believe this concern may stem from a misunderstanding. Although we clarified this in the current version of the paper, we will further refine the manuscript to make this point more evident.

---

> ### Author Response · Authors · 2025-11-17
> **Response to the Weakness 2**
>
> We thank the reviewer for raising this concern. We believe the issue arises from a misunderstanding of the locality design in SAKE.
>
> First, in Figure 4, the emotion “sad” is the edit target rather than the original pre-edit label. This is indicated in the figure.
>
> Second, the locality metric does not evaluate the correctness of the locality sample. As defined in Equation (3), locality measures whether the model preserves its original behavior before the edit. The goal is to ensure that knowledge unrelated to the edit remains unchanged. The expected output for a locality sample is therefore the model’s pre-edit prediction rather than the ground-truth label. In Figure 4, the model predicts “sad” for Audio Locality Type 2 even though the ground truth is “angry”. This is acceptable because the locality metric only examines whether the same prediction is preserved after the edit. This explanation is consistent with the discussion beginning at line 170 in the submitted manuscript.
>
> Third, for Audio Locality Type 3, the intention is to ensure that the model’s original knowledge of the edit target is not unintentionally altered. When editing from one emotion to another, it is crucial that the model’s original understanding of the target emotion remains intact. This rationale is explained around line 239 of the manuscript. In Figure 4, the model’s prediction for Audio Locality Type 3 is already correct before the edit, and the fact that it continues to output the edit target after the edit demonstrates that this original knowledge is successfully preserved. The use of “sad” in the locality sample is therefore a deliberate design choice to verify that the model maintains its understanding of the edit target.
>
> More broadly, even if the model makes an incorrect prediction before the edit, preserving that incorrect prediction after the edit still constitutes good locality. In contrast, if the model suddenly becomes correct because of the propagation of the edit target, as suggested by the reviewer, this is treated as a failure in locality. The metric explicitly captures this situation.
> Finally, the comment overlooks the fact that SAKE evaluates locality through multiple complementary types. These types capture knowledge outside the edit scope at different levels. An edit that merely propagates the target label cannot achieve strong performance across all locality types. The benchmark, therefore, reliably distinguishes preservation of unrelated knowledge from unintended propagation of edited knowledge.
>
> We hope this clarifies the misunderstanding. We appreciate the reviewer’s attention and will refine the manuscript to make these points even clearer.

---

### Comment · Area_Chair_5yv2 · 2025-11-27
**Please review the authors' responses and provide feedback ASAP**

Dear Reviewers,

Thank you for your essential contributions to the review process. The authors have submitted their responses to your initial reviews.

I kindly ask you to carefully review the authors' responses for this submission. Your timely assessment of how the authors have addressed your original concerns is a critical step in reaching a final decision.

Please provide your feedback and any necessary updates to your reviews as soon as possible to ensure we can meet our tight schedule for the discussion phase.

Your prompt attention to this matter is highly appreciated.

Regards,

-AC

---

### Author Response · Authors · 2025-12-02
**Summary of the Rebuttal**

We appreciate all the reviewers’ feedback. We have addressed every weakness and question raised during the rebuttal process and updated the manuscript accordingly, with all revisions highlighted in blue. This comment summarizes the key outcomes of the discussion period.

## Our Major Contribution

Reviewers generally agree that our paper explores a novel and impactful research direction. To the best of our knowledge, this is the first work to investigate knowledge editing in large audio-language models (LALMs) for auditory attributes, a direction with substantial potential applications. We introduce SAKE, the first benchmark specifically designed for auditory knowledge editing, featuring a comprehensive evaluation framework across four dimensions. The novelty, rigor, and significance of this benchmark were explicitly recognized by several reviewers (vcdK, gqjC, c6Zy).

Beyond benchmark construction, our experimental results reveal important challenges faced by current knowledge editing methods. For instance, we find that preserving intra-attribute auditory knowledge during editing is far more difficult than expected. This highlights the inherent difficulty of editing auditory attributes and provides insights that can guide future advancements. These findings were also acknowledged and appreciated by reviewers (gqjC, c6Zy).

In summary, our contributions are two-fold:
1. We construct the first systematic benchmark for auditory knowledge editing in LALMs, an unexplored yet crucial direction that ensures strong novelty and impact.


2. We reveal key challenges of auditory knowledge editing through rigorous experiments, offering actionable insights for future method development.


Overall, this work opens up a new research area with significant implications for the audio processing community.

## Improvements During Discussion Period

We have made several improvements to the paper based on the reviewers’ comments.

First, we elaborate on the rationale behind our benchmark design, including the number of covered attributes, as well as the experimental setup concerning the choice of edited models. We clarify why these choices are appropriate and consistent with common practice. The detailed explanation is now provided in **Appendix C** of the revised manuscript.

Second, following Reviewer vcdK’s suggestion, we include an additional editing method (i.e., WISE) as a new baseline. Although the original paper already evaluated seven editing methods, adding WISE further strengthens the completeness of our comparison. The results of this newly evaluated method are included in **Sec. 5**, with full results available in **Appendix F** and **Appendix G**, and implementation details in **Appendix D**.

Third, in response to Question 2 from Reviewer c6Zy, we expand our discussion on **intra-attribute knowledge entanglement**, offering a deeper explanation of the potential causes of this phenomenon. This enhanced discussion appears in **Appendix H**.

Fourth, responding to Question 3 from Reviewer c6Zy, we provide a more fine-grained breakdown of the **portability evaluation**, allowing a clearer examination of how related knowledge is, or is not, successfully updated during editing. The detailed analysis is presented in **Appendix I**.

Finally, we note that several concerns raised by the reviewers stem from misunderstandings of our work. We address these issues directly in our responses and refine the manuscript where necessary to reduce the likelihood of similar misunderstandings.

In summary, through detailed clarifications, additional experiments, and expanded analyses, we believe we have thoroughly addressed all concerns raised by the reviewers and substantially improved the quality of the paper. Once again, we sincerely thank the reviewers for their time and dedication, which have helped us improve this work.

---

### Meta-Review · Area_Chair_JjUM · 2025-12-28

**Summary:**

SAKE proposes a benchmark for auditory attribute knowledge editing in speech-to-text LALMs, covering four attributes (gender, emotion, language, animal sound) and evaluating editing along reliability, generality, audio/text locality, portability, plus sequential editing. Reviewers agree the problem is novel and the benchmark design (especially intra-attribute locality and portability) is well-motivated. The rebuttal clarified several misunderstandings (e.g., reliability vs. generality scope, and locality measuring preservation of pre-edit behavior) and added one additional baseline (WISE) plus extra analyses (entanglement and portability breakdown).

However, the key reasons for recommending Reject remain: as a benchmark paper, the evaluation breadth is still limited (2 LALMs, 4 attributes, and incomplete SOTA coverage beyond the added WISE), and some core concerns persist around whether the “continuous/abstract” motivation is fully realized (evaluation is largely via discrete labels) and whether the diagnostic/mechanistic analysis is deep enough (e.g., why IKE variants fail, and stronger interpretability/representation-level evidence). Overall, despite improvements, the paper’s scope and depth fall slightly below the main-conference acceptance bar.

**Reviewer Concerns:**

Addressed by the rebuttal / revision
- Edit scope clarity (instance vs. generalization) (J6YR): clarified that edits are applied on single instances; generalization is explicitly evaluated via the generality track.
- Locality “overlap” confusion (J6YR): clarified locality is consistency with pre-edit outputs, not correctness; Type-3 locality intentionally checks preserving the model’s original knowledge of the target label.
- Paraphrase quality control (gqjC): confirmed paraphrases are manually verified.
- Audio generation critique is out-of-scope (vcdK): benchmark targets speech-to-text LALMs (text output only).
- Additional baseline (vcdK): WISE added; results discussed in the revision.

Still outstanding (or partially addressed)
- Limited method coverage (vcdK): WISE added, but other suggested SOTA baselines remain unevaluated; breadth is still limited for a benchmark positioning.
- Attribute breadth & “continuous/abstract” gap (c6Zy, gqjC, vcdK): authors clarify “continuous” refers to acoustic variability, but benchmark evaluation still largely operates via discrete labels and only four attributes.
- Limited model diversity (2 LALMs) (gqjC, c6Zy): authors justify representativeness, but evidence remains limited for generality across architectures.
- Mechanistic/interpretability depth (vcdK): authors note interpretability is hard and defer; still, lack of deeper diagnostics weakens the benchmark insight.
- IKE failure mechanism (c6Zy): authors clarify they tested instruction-only (I-IKE) vs instruction+example (IE-IKE), but deeper isolation/diagnosis remains limited.

**Reviewer Scores:**

**J6YR**: possibly 4 → 6. Their two main issues were clarifications (scope/locality), largely resolved by rebuttal.

**vcdK**: likely stays 2 (at best 2→4). Adding WISE helps, but core concerns (breadth + mechanistic depth) remain.

**gqjC**: likely stays 4 (possibly 4→5). Paraphrase check and clarifications help; limited models/attributes remain.

**c6Zy**: likely stays 6. Some questions were addressed (IKE variants, entanglement discussion, portability breakdown), but scope mismatch concerns may persist.

---

### Decision · Program_Chairs · 2026-01-26

Reject